

# Radiative effects of inter-annually varying versus inter-annually invariant aerosol emissions from fires

Benjamin S. Grandey[1], Hsiang-He Lee[1], and Chien Wang[2,1]

[1]Center for Environmental Sensing and Modeling, Singapore-MIT Alliance for Research and Technology, Singapore.
[2]Center for Global Change Science, Massachusetts Institute of Technology, Cambridge, Massachusetts, USA.

*Correspondence to:* Benjamin S. Grandey (benjamin@smart.mit.edu)

**Abstract.** Open-burning fires play an important role in the Earth's climate system. In addition to contributing a substantial fraction of global emissions of carbon dioxide, they are also a major source of atmospheric aerosols such as organic carbon, black carbon, and sulphate. These "fire aerosols" can influence the climate via both direct and indirect radiative effects. In this study, we investigate these radiative effects and the hydrological fast response using the Community Atmosphere Model version 5

(CAM5). Emissions of fire aerosols exert a global mean net radiative effect of $-1.0\,\mathrm{W\,m^{-2}}$, dominated by the cloud shortwave response to organic carbon aerosol. The net radiative effect is particularly strong over boreal regions. By comparing simulations using inter-annually varying versus inter-annually invariant emissions, we find that ignoring the inter-annual variability of the emissions can lead to systematic overestimation of the strength of the net radiative effect of the fire aerosols. Globally, the overestimation is $+23\,\%$ ($-0.2\,\mathrm{W\,m^{-2}}$). Regionally, the overestimation can be substantially larger. For example, over Australia and

New Zealand the overestimation is $+58\,\%$ ($-1.2\,\mathrm{W\,m^{-2}}$), while over Boreal Asia the overestimation is $+43\,\%$ ($-1.9\,\mathrm{W\,m^{-2}}$). The systematic overestimation of the net radiative effect of the fire aerosols is likely due to the non-linear influence of aerosols on clouds. However, ignoring inter-annual variability in the emissions does not appear to significantly impact the hydrological fast response. In order to improve understanding of the climate system, we need to more accurately quantify the effects of aerosols, taking into account important characteristics such as inter-annual variability.

# 1  Introduction

There are many types of open-burning fires, caused by both natural and human influences. Broad categories include agricultural waste burning, grassland fire, peat fire, and various types of forest fire (van der Werf et al., 2010). In addition to producing heat, these fires emit pollutants into the atmosphere. One such pollutant is carbon dioxide, contributing to climate change – Page et al. (2002) estimate that the large-scale burning of peat and forests in 1997 in Indonesia alone emitted the equivalent of 13–40 %

of global annual emissions of carbon dioxide from fossil fuels. Other pollutants include aerosols, containing organic carbon and black carbon. Some fires, including burning of sulphur-containing peat in Indonesia (Gras et al., 1999; Balasubramanian et al., 1999; Langmann, 2003), also emit sulphur dioxide, leading to the formation of sulphate aerosol. These aerosols have a negative impact on air quality and human health (Lelieveld et al., 2015).





Aerosols also affect climate both regionally and globally. Aerosols scatter or absorb incoming sunlight, cooling the earth's surface, an effect known as the "direct aerosol effect" (Haywood and Boucher, 2000). Absorbing aerosols, such as black carbon, warm the atmosphere, potentially affecting cloud formation (the "semi-direct effect; Ackerman, 2000). Hygroscopic aerosols, such as sulphate, also play an important role in cloud formation, acting as the cloud condensation nuclei on which

cloud droplets condense. Hence changing the availability of sulphate aerosol may affect the droplet size and number in clouds, potentially changing cloud reflectivity (the "cloud albedo effect" or "first indirect effect"; Twomey, 1974, 1977), or cloud lifetime (the "cloud lifetime effect" or "second indirect effect"; Albrecht, 1989). In addition to these two indirect effects, many other aerosol effects on clouds have also been proposed (Tao et al., 2012; Rosenfeld et al., 2014).

Surface cooling and atmospheric heating perturb regional temperature gradients, potentially affecting large-scale circulations

such as monsoon systems. For example, many studies have suggested that aerosols impact the South Asian monsoon (e.g. Chung et al., 2002; Lau et al., 2006; Wang et al., 2009; Bollasina et al., 2011; Lee et al., 2013; Lee and Wang, 2015). Remote impacts are also possible. For example, we have recently demonstrated that anthropogenic emissions of aerosols from Asia may affect rainfall in remote locations such as Australia and the Sahel (Grandey et al., 2016). Remote impacts have also been explored by Menon (2002), Wang (2007, 2009), and Teng et al. (2012).

Most studies investigating the climate impacts of aerosols have focused on anthropogenic aerosols emitted by industry or power plants. Fewer studies have focused on the radiative and climate impacts of aerosols emitted from open-burning fires, which we will refer to as "fire aerosols", the focus of our current study.

Clark et al. (2015) recently investigated the radiative and climate effects of sub-monthly variability in fire emissions. They found that the "monthly-mean emissions approximation holds roughly in the tropics, where fires are more frequent and less

episodic; however it does not perform as well in the boreal regions" (Clark et al., 2015). Jeong and Wang (2010) demonstrated the importance of seasonal variation in the emissions of fire aerosols. They found that the "seasonality of biomass burning emissions uniquely affects the global distributions of convective clouds and precipitation" (Jeong and Wang, 2010). In light of the findings of Jeong and Wang (2010), we use seasonally varying emissions in our study. Beyond seasonal variability, Feng and Christopher (2014) and Sena and Artaxo (2015) showed that the direct radiative effects of fire aerosols also exhibit inter-

annual variability, something that is not taken into account in most climate modelling studies. We complement the sub-monthly focus of Clark et al. (2015) and the seasonal focus of Jeong and Wang (2010) by investigating inter-annually varying emissions.

Three of the above-mentioned studies (Jeong and Wang, 2010; Feng and Christopher, 2014; Sena and Artaxo, 2015) considered the direct radiative effects of fire aerosols but did not consider other radiative effects. However, as pointed out by Jacobson (2014), the radiative effects of fire aerosols are not limited to the direct effect. Jiang et al. (2016) found that the indirect effects

of fire aerosols may be substantially larger than the direct effect. In addition to representing the direct and semi-direct effects, the aerosol–climate model used in this study also includes a representation of indirect effects on stratiform clouds. However, some other possible effects, such as cloud absorption effects (Jacobson, 2014), are not represented.

Using the ECHAM6-HAM2 aerosol-climate model, Veira et al. (2015) recently investigated the influence of fire emission height. Although they found that emission height can influence the radiative effects of fire aerosols, they concluded that "Sig-

nificant improvements in aerosol wildfire modeling likely depend on better emission inventories and aerosol process modeling





rather than on improved emission height parametrizations" (Veira et al., 2015). In this study, we prescribe emission heights according to fire type, as described in the Method section.

Building on these previous studies, we explore the radiative effects associated with inter-annually varying emissions of fire aerosols. We use Global Fire Emissions Database (GFED) aerosol emissions to drive a global aerosol–climate model. We

quantify the radiative effects for the globe and several different regions (Fig. 1). We also discuss the hydrological fast response. In the Discussion section, we focus on one primary research question: how do the effects of inter-annually varying emissions differ from those of inter-annually invariant emissions? In the context of this research question, we consider the non-linear influence of aerosols on clouds.

## 2 Method

### 2.1 Model configuration

The Community Earth System Model (CESM) version 1.2.2, which includes the Community Atmosphere Model version 5 (CAM5) (Neale and Coauthors, 2012), is used. CAM5 contains a modal aerosol model with three lognormal modes (MAM3) (Liu et al., 2012). Aerosol indirect effects on stratiform clouds are represented via coupling between the aerosols and the stratiform cloud microphysics (Morrison and Gettelman, 2008; Gettelman et al., 2010). As a result of these indirect effects

(Ghan et al., 2012), CESM1-CAM5 produces a relatively strong total aerosol radiative effect compared to many other global climate models (Table 7 of Shindell et al., 2013).

A finite volume grid with a horizontal resolution of approximately $1.9° × 2.5°$ and 30 levels is used for CAM5. The land model is run at the same horizontal resolution. Greenhouse gas concentrations, sea-surface temperatures (SSTs), and sea-ice are prescribed using year-2000 climatological values.

The "prescribed-SST" approach is suitable for diagnosing radiative flux perturbations (RFPs) (Haywood et al., 2009) and the hydrological "fast response" (Bala et al., 2010) associated with aerosols. RFPs are diagnosed by calculating the difference in the top-of-atmosphere radiative flux between two prescribed-SST simulations forced with different aerosol emissions. In contrast to a strict definition of "radiative forcing", the RFP approach allows clouds and precipitation to adjust to the aerosol forcing via fast feedback processes. Hence RFPs allow quantification of the indirect effects of aerosols on clouds. To facilitate

decomposition of the RFPs (Ghan, 2013), the radiation scheme is called twice at each radiation time step. The first radiation call includes all aerosol species. The second radiation call, which is purely diagnostic, excludes all aerosol species, allowing diagnosis of "clean-sky" fluxes.

### 2.2 Emissions

Emissions of organic carbon, black carbon, sulphur dioxide, primary sulphate, dimethyl sulphide, and secondary organic

aerosol precursors mostly follow the default MAM3 emissions for year-2000. The exception is that fire emissions of organic carbon, black carbon, sulphur dioxide, and primary sulphate are modified.



As in the "interpolation method" simulation of Clark et al. (2015), monthly fire emissions are used in this study. The monthly fire emissions are obtained from the Global Fire Emission Database version 4.0 with small fires included (GFED4.0s) (revised version of van der Werf et al., 2010). The recommended GFED4.0s emission factors are used to convert partitioned dry matter emissions to emissions of organic carbon, black carbon, and sulphur dioxide. These emissions are then conservatively remapped

from the GFED4.0s grid ($0.25° \times 0.25°$) to the grid used in the MAM3 emissions files (approximately $1.9° \times 2.5°$).

Following Liu et al. (2012), emission height profiles are based on Table 4 of Dentener et al. (2006). For boreal forest fires, the "Boreal (Eurasia)" emission height profile is used instead of the "Boreal (Canada)" profile. The Dentener et al. (2006) height profiles are linearly interpolated to the higher resolution vertical levels used in the MAM3 emission files.

The GFED4.0s fire categories are subsequently mapped to the MAM3 emission sector categories. The GFED4.0s "agri-

cultural waste burning" category corresponds to the MAM3 agricultural waste burning category; the GFED4.0s "savanna, grassland, and shrubland fires" category corresponds to the MAM3 grass fire category; and the GFED4.0s "boreal forest fires", "temperate forest fires", "tropical forest fires (deforestation and degradation)", and "peat fires" categories are combined into the MAM3 forest fire category. The resulting global and regional organic carbon, black carbon, and sulphur dioxide annual emissions for the different simulations (described below) are shown in Fig. 2 and supplemental material Figs. S1–S5. Glob-

ally, averaged across 1997–2006, fire emissions are responsible for 60 % ($18 \, \text{Tg yr}^{-1}$) of the total organic carbon emissions of $30 \, \text{Tg yr}^{-1}$, 28 % ($2 \, \text{Tg yr}^{-1}$) of the total black carbon emissions of $7 \, \text{Tg yr}^{-1}$, and 2 % ($2.6 \, \text{Tg yr}^{-1}$) of the total sulphur dioxide emissions of $131 \, \text{Tg yr}^{-1}$ – it is worth noting that the total black carbon emissions of $7 \, \text{Tg yr}^{-1}$ used in this study may represent an underestimate. Cohen and Wang (2014) estimate that global BC emissions are $17.8 \pm 5.6 \, \text{Tg yr}^{-1}$.

It is assumed that 2.5 % of the sulphur dioxide is emitted as primary sulphate (Dentener et al., 2006; Liu et al., 2012).

Size distributions of emitted organic carbon, black carbon, and primary sulphate follow the size distributions described in the supplementary material of Liu et al. (2012).

## 2.3   Simulations

Fifteen simulations are performed. The first of these, F0, is a control simulation. In F0, fire emissions of organic carbon, black carbon, sulphur dioxide, and primary sulphate are set to zero. In this paper, RFPs are calculated with respect to simulation F0,

rather than a pre-industrial control.

Ten simulations use fire emissions for the ten different years between 1997 and 2006. For example, F1997 uses year-1997 fire emissions, F1998 uses year-1998 fire emissions etc. Together, this ensemble of ten simulations is referred to as {Fyyyy}. Comparison of {Fyyyy} with F0 reveals the radiative effect of inter-annually varying emissions of fire aerosols. We refer to this as the "revised" approach, in contrast to the "conventional" approach described in the next paragraph.

In contrast to the {Fyyyy} ensemble, the "FMEAN" simulation uses mean emissions averaged across 1997–2006 for each month. The seasonal cycle is retained, but inter-annual variability is removed. We refer to this as the "conventional" approach. Comparison of FMEAN ("conventional") with {Fyyyy} ("revised") reveals the influence of ignoring inter-annual variability.

"OMEAN" combines the organic carbon emissions of FMEAN with the black carbon, sulphur dioxide, and primary sulphate emissions of F0 (i.e. zero emissions of black carbon and sulphur from fires). Similarly, "BMEAN" combines the black carbon



emissions of FMEAN with the organic carbon, sulphur dioxide, and primary sulphate emissions of F0; and "SMEAN" combines the sulphur dioxide and primary sulphate emissions of FMEAN with the black carbon and organic carbon emissions of F0. Comparison of OMEAN, BMEAN, and SMEAN with FMEAN reveals the relative contributions of organic carbon, black carbon, and sulphur dioxide.

The first two years of each simulation are treated as spin-up. Simulations F0 and FMEAN are each run for 44 years, providing an analysis period of 42 years. Simulations OMEAN, BMEAN, SMEAN, and each member of the {Fyyyy} ensemble (i.e. F1997 etc.) are each run for fourteen years, providing an analysis period of twelve years. The total analysis period for {Fyyyy} is 120 years, because {Fyyyy} consists of ten separate simulations. When deciding the simulation lengths, we sought a balance between the improved statistical power of larger sample sizes and the computational expense of the simulations.

## 3    Results

### 3.1    Radiative effects of fire aerosols

#### 3.1.1    Global mean

In most global climate modelling studies, the inter-annual variability of aerosol emissions from fires is ignored. Here, we also start with this "conventional" approach of ignoring inter-annual variability. The "conventional" global mean net RFP associated

with fire aerosols is $-1.3\,\mathrm{W\,m^{-2}}$ (Table 1), comparable to that found by Clark et al. (2015, their "interpolation method" simulation). This is smaller than CESM-CAM5's anthropogenic aerosol year-2000−year-1850 net RFP of $-1.5\,\mathrm{W\,m^{-2}}$ (Ghan et al., 2012), but much larger than ECHAM6-HAM2's fire aerosol net RFP of $-0.2\,\mathrm{W\,m^{-2}}$ (Veira et al., 2015). CESM-CAM5 is known to produce a strong net RFP compared to many other global climate models (Shindell et al., 2013).

This global mean net RFP of $-1.3\,\mathrm{W\,m^{-2}}$ is dominated by the cloud shortwave RFP of $-1.1\,\mathrm{W\,m^{-2}}$ (Fig. 3a, Table 2),

primarily driven by the organic carbon emissions (Fig. 3a), in general agreement with the findings of Jiang et al. (2016). The cloud longwave RFP of $-0.1\,\mathrm{W\,m^{-2}}$ (Fig. 3a, Table S1) is also driven by the organic carbon emissions (Fig. 3a). Interestingly, again in agreement with Jiang et al. (2016), the cloud longwave RFP associated with the fire aerosols is negative, in contrast to the results of Ghan et al. (2012) who found a positive cloud longwave RFP associated with anthropogenic aerosols (see also Gettelman et al., 2012). The occurrence of negative cloud longwave RFP values corresponds to decreasing fractional cover of

high clouds and decreasing ice water path, especially over equatorial land regions (Fig. S11).

The organic carbon emissions drive a small surface albedo RFP of $-0.1\,\mathrm{W\,m^{-2}}$ (Fig. 4a, Table S2), due to increasing snow cover in Boreal North America (see below) and other parts of the world, likely in response to the cloud shortwave RFP. The organic carbon emissions also drive a small direct effect RFP, but this is offset by the positive direct effect RFP of the black carbon emissions (Fig. 4a).

When the emissions for individual years are used to drive separate simulations in the ten-member {Fyyyy} ensemble, the inter-annual variability of the RFP components is revealed. Most of the variability in global mean net RFP is dominated by the cloud shortwave RFP component (Fig. 3a). The global mean net RFP ranges from $-0.8\,\mathrm{W\,m^{-2}}$ in F2000, which has the lowest





organic carbon emissions, to $-1.3\,\mathrm{W\,m^{-2}}$ in F1998, which has the second highest organic carbon emissions (Figs. 2a and 3a). Interestingly, this maximum net RFP of $-1.3\,\mathrm{W\,m^{-2}}$ in F1998 is comparable to the "conventional" global mean net RFP, suggesting that the global impact of the fire aerosols may saturate (see Discussion). The simulation with the highest organic carbon emissions (Fig. 2a), F1997, has a comparatively weak global mean net RFP of $-0.9\,\mathrm{W\,m^{-2}}$ (Fig. 3a), demonstrating

that the global mean net RFP response is not a monotonic function of global annual organic carbon emissions (Fig. 5a). This suggests that the location and timing of emissions play an important role in determining the RFP response.

In contrast to the "conventional" approach of using inter-annually invariant emissions, comparison of {Fyyyy} with F0 reveals the mean effect of inter-annually varying emissions of fire aerosols. The "revised" global mean net RFP is $-1.0\,\mathrm{W\,m^{-2}}$ (Table 1). A statistically significant "conventional"$-$"revised" difference of $-0.24\,\mathrm{W\,m^{-2}}$ is found (Table 1), primarily due to

the cloud shortwave RFP component (Fig. 3a). This shows that the conventional approach of using inter-annually invariant fire aerosol emissions leads to a 23 % overestimation of the negative RFP exerted by fire aerosols on the climate system (Fig. 5a). Further discussion of the effect of ignoring inter-annual variability can be found in the Discussion section.

### 3.1.2 Global distribution

The global distribution of the radiative effects associated with fire aerosols is highly inhomogeneous (Fig. 6a,b). Particularly

strong net RFPs occur over high latitude boreal regions, and parts of South America, Southern Hemisphere Africa, Australia, and the Maritime Continent. These regions will be discussed below. The radiative effects of the fire aerosols are not limited to land regions, but extend over ocean regions downwind of fire sources. Net RFPs stronger than $-5\,\mathrm{W\,m^{-2}}$ occur over parts of the North Pacific, Tropical Pacific, and Tropical Atlantic oceans (Fig. 6a,b).

Organic carbon emissions are almost entirely responsible for the negative net RFPs associated with fire aerosols (Fig. S9a).

By comparison, the contributions of black carbon and sulphur dioxide are much smaller (Fig. S9b,c).

### 3.1.3 Boreal North America and Boreal Asia

Forest fires dominate the fire aerosol emissions from both Boreal North America (Fig. 2b) and Boreal Asia (Fig. 2c). These boreal regions are prone to the strongest net RFPs of any of the land regions (Table 1) and are therefore discussed first.

Boreal North America has a "conventional" net RFP of $-6.5\,\mathrm{W\,m^{-2}}$ (Table 1). This net RFP is dominated by the cloud

shortwave RFP of $-5.9\,\mathrm{W\,m^{-2}}$ (Table 2), driven by the organic carbon emissions (Fig. 3b). As pointed out by Jiang et al. (2016), "the large cloud liquid water path over land areas of the Arctic favors the strong fire aerosol indirect effect" (Jiang et al., 2016). The next largest component is the surface albedo RFP of $-0.6\,\mathrm{W\,m^{-2}}$ (Table S2), driven by both organic carbon and sulphur emissions (Fig. 4b), reflecting an increase in snow cover. In comparison, the cloud longwave RFP (Fig. 3b, Table S1) and direct effect RFP (Fig. 4b, Table S3) are much smaller.

Compared to Boreal North America, Boreal Asia has a slightly weaker "conventional" net RFP of $-6.2\,\mathrm{W\,m^{-2}}$ (Table 1). As was the case for Boreal North America, the net RFP for Boreal Asia is dominated the cloud shortwave RFP of $-5.8\,\mathrm{W\,m^{-2}}$ (Table 2), driven by the organic carbon emissions (Fig. 3c). The surface albedo RFP of $-0.3\,\mathrm{W\,m^{-2}}$ (Table S2) and cloud




longwave RFP of $-0.2\,\mathrm{W\,m^{-2}}$ (Table S1) are also driven by the organic carbon emissions (Figs. 4c and 3c). The direct effect RFP is much smaller (Table S3).

For both boreal regions, the large inter-annual variability of organic carbon emissions contributes to large variability in net RFP (Fig. 5b,c). Over Boreal North America, a strong correlation of $-0.83$ between the annual organic carbon emissions and the net RFP demonstrates that the annual total organic carbon emissions can explain $69\,\%$ of the variance in net RFP (assuming a linear relationship). F2001, which has the lowest emissions, has a net RFP of $-2.5\,\mathrm{W\,m^{-2}}$; while F2004, which has the highest emissions, has a net RFP of $-6.9\,\mathrm{W\,m^{-2}}$ (Figs. 3b and 5b). This variability in the net RFP is dominated by variability in the cloud shortwave RFP (Fig. 3b).

Over Boreal Asia, a correlation of $-0.70$ demonstrates that the variability in the annual total organic carbon emissions can explain $49\,\%$ of the variance in net RFP (Fig. 5c). F2004, which has the lowest emissions, has a net RFP of $-2.7\,\mathrm{W\,m^{-2}}$; while F2003, which has the highest emissions, has a net RFP of $-5.1\,\mathrm{W\,m^{-2}}$ (Figs. 3c and 5c). F1998, which has much lower emissions than F2003, has an even stronger net RFP of $-6.0\,\mathrm{W\,m^{-2}}$, indicating that the regional annual total emissions are not the sole determining factor of net RFP strength. This suggests that variability in timing and the specific location of the fires also plays a role, possibly due to the non-linear influence of aerosols on clouds (see Discussion).

When the inter-annually variability of emissions is taken into account, the "revised" net RFPs are significantly weaker than the "conventional" net RFPs over both boreal regions. Over Boreal North America, the "conventional" approach of ignoring inter-annual variability leads to a $32\,\%$ overestimation of net RFP strength (Fig. 5b, Table 1) and a $33\,\%$ overestimation of cloud shortwave RFP strength (Table 2). Over Boreal Asia, the "conventional" approach leads to a $43\,\%$ overestimation of net RFP strength (Fig. 5c, Table 1) and a $47\,\%$ overestimation of cloud shortwave RFP strength (Table 2).

### 3.1.4 Northern Hemisphere South America and Southern Hemisphere South America

Both grass fires and forest fires substantially contribute to the fire aerosol emissions from both South American regions (Fig. 2d,h). After the boreal regions, Northern Hemisphere South America has the next largest "conventional" net RFP (Table 1). In contrast to the boreal regions, the net RFP of $-3.8\,\mathrm{W\,m^{-2}}$ is dominated by the cloud longwave RFP of $-2.6\,\mathrm{W\,m^{-2}}$ (Fig. 3d, Table S1), driven by the organic carbon emissions (Fig. 3d). The negative cloud longwave RFP is associated with a decrease in high cloud fraction and a decrease in ice water path (Fig. S11).

The cloud shortwave RFP of $-1.1\,\mathrm{W\,m^{-2}}$ (Table 2) also contributes to the net RFP. Interestingly, the black carbon emissions cause a substantial negative cloud shortwave RFP, although this cloud shortwave RFP is weaker than that of the organic carbon emissions (Fig. 3d). The direct effect RFP and surface albedo RFP are much smaller (Fig. 4d, Tables S2 and S3).

In contrast to Northern Hemisphere South America, Southern Hemisphere South America has a weaker "conventional" net RFP is $-1.8\,\mathrm{W\,m^{-2}}$ (Table 1), with approximately equal contributions of $-0.8\,\mathrm{W\,m^{-2}}$ coming from the cloud shortwave RFP and the cloud longwave RFP (Fig. 3h, Tables 2 and S1). Both the cloud shortwave RFP and the cloud longwave RFP are primarily driven by the organic carbon emissions (Fig. 3h).

Net RFP exhibits inter-annual variability over both South American regions. For example, over Northern Hemisphere South America, F2006, which has the lowest emissions, has a net RFP of $-2.7\,\mathrm{W\,m^{-2}}$; while F2003, which has the highest emissions,




has a net RFP of $-4.7\,\mathrm{W\,m^{-2}}$ (Figs. 3d and 5d). Variability in the annual total organic carbon emissions can explain $59\,\%$ of the variance in net RFP over Northern Hemisphere South America (Fig. 5d) and $74\,\%$ of the variance in net RFP over Southern Hemisphere South America (Fig. 5h).

For both South American regions, the "revised" net RFP is comparable to the "conventional" net RFP (Table 1). In fact, the "conventional"$-$"revised" net RFP differences are statistically insignificant, indicating that the "conventional" approach of ignoring inter-annual variability does not significantly impact net RFP over either South American region.

### 3.1.5 Australia and New Zealand

As was the case for the South American regions, both grass fires and forest fires contribute to the fire aerosol emissions from the "Australia and New Zealand" region (Fig. 2e). The "conventional" net RFP is $-3.3\,\mathrm{W\,m^{-2}}$ (Table 1). As was the case over the Boreal regions, the net RFP over Australia and New Zealand is dominated the cloud shortwave RFP (Table 2), driven by the organic carbon emissions (Fig. 3e). The cloud longwave RFP, surface albedo effect RFP, and direct effect RFP are all statistically insignificant (Tables S1–S3).

Only a weak correlation exists between annual total organic emissions and net RFP over Australia and New Zealand (Fig. 5e). Of the {Fyyyy} ensemble members, F2003 has the highest organic carbon emissions (Fig. 2e) yet has the weakest net RFP of $-0.9\,\mathrm{W\,m^{-2}}$ (Figs. 3e and 5e). F2002, which has comparable organic carbon emissions, has a much larger net RFP of $-4.1\,\mathrm{W\,m^{-2}}$. It is clear that the annual total emission of organic carbon is not the primary driver of inter-annual variability in net RFP over Australia and New Zealand.

The "revised" net RFP of $-2.1\,\mathrm{W\,m^{-2}}$ is significantly weaker that the "conventional" net RFP of $-3.3\,\mathrm{W\,m^{-2}}$ (Table 1). The "conventional" approach overestimates the strength of net RFP by $58\,\%$ (Fig. 5e), the largest percentage difference found for any of the regions discussed in this section.

### 3.1.6 Southern Hemisphere Africa

In contrast to all the regions discussed above, the fire aerosol emissions from Southern Hemisphere Africa are almost completely driven by grass fires (Fig. 2f). The "conventional" net RFP is $-3.3\,\mathrm{W\,m^{-2}}$ (Table 1). The cloud shortwave RFP of $-2.0\,\mathrm{W\,m^{-2}}$ (Table 2) and the cloud longwave RFP of $-1.4\,\mathrm{W\,m^{-2}}$ (Table S1) are both driven by the organic carbon emissions (Fig. 3f).

Net RFP ranges from $-2.2\,\mathrm{W\,m^{-2}}$ in F1999 to $-3.7\,\mathrm{W\,m^{-2}}$ in F1997, F2000, and F2005 (Fig. 3f). The correlation between annual total organic carbon emissions and net RFP is very weak (Fig. 5e). Variability in the annual total organic carbon emissions can explain less than $3\,\%$ of the variance in net RFP. Other factors, such as the specific timing and location of emissions, must drive the inter-annual variability in net RFP.

The "revised" net RFP of $-3.2\,\mathrm{W\,m^{-2}}$ is comparable to the "conventional" net RFP of $-3.3\,\mathrm{W\,m^{-2}}$ (Table 1). The difference is statistically insignificant. This indicates that the "conventional" approach does not significantly impact net RFP over Southern Hemisphere Africa.



### 3.1.7 Western Maritime Continent

Forest fires dominate the fire aerosol emissions from the Western Maritime Continent (Fig. 2g). The "conventional" net RFP is $-2.5\,\mathrm{W\,m^{-2}}$ (Table 1). In contrast to the regions discussed above, with the exception of Northern Hemisphere South America, the net RFP is dominated by the cloud longwave RFP of $-1.7\,\mathrm{W\,m^{-2}}$ (Fig. 3g, Table S1), driven by the organic carbon emissions (Fig. 3g). The cloud shortwave RFP is actually statistically insignificant (Table 2).

A relatively strong correlation exists between annual organic carbon emissions and net RFP, with variability in annual organic carbon emissions able to explain $53\,\%$ of the variance in net RFP (Fig. 5g). One outlier likely contributes disproportionately to the correlation: F1997 has by far the highest fire aerosol emissions (Fig. 2g) also has the largest net RFP of $-3.0\,\mathrm{W\,m^{-2}}$ (Fig. 3g). Among the other {Fyyyy} ensemble members, the net RFP varies between $-0.4\,\mathrm{W\,m^{-2}}$ and $-2.2\,\mathrm{W\,m^{-2}}$. The relationship between organic carbon emissions and net RFP appears to be non-linear (Fig. 5g).

The "revised" net RFP of $-1.6\,\mathrm{W\,m^{-2}}$ is significantly weaker that the "conventional" net RFP of $-2.5\,\mathrm{W\,m^{-2}}$ (Table 1). The "conventional" approach overestimates the strength of net RFP by $49\,\%$ (Fig. 5g). Interestingly, although the net RFP is dominated by the cloud longwave RFP, the "conventional"−"revised" difference in cloud longwave RFP is relatively small ($-0.2\,\mathrm{W\,m^{-2}}$) and statistically insignificant (Table S1). The "conventional"−"revised" difference in cloud shortwave RFP is larger ($-0.6\,\mathrm{W\,m^{-2}}$), although this is also statistically insignificant (Table 2). These statistically insignificant differences in cloud longwave RFP and cloud shortwave RFP combine to form a statistically significant difference in net RFP.

### 3.1.8 Other land regions

Six of the other eight land regions have statistically significant "conventional" net RFPs ranging from $-0.6\,\mathrm{W\,m^{-2}}$ to $-1.6\,\mathrm{W\,m^{-2}}$ (Table 1). The "conventional" approach leads to a statistically significant overestimation of the strength of the net RFP over only two of these six regions (Table 1).

## 3.2 The hydrological fast response

In addition to facilitating the calculation of RFPs, the prescribed-SST simulations analysed in this paper facilitate investigation of the hydrological fast response (Bala et al., 2010). The hydrological fast response to the fire aerosols is discussed here; investigation of the hydrological slow response, which depends on the ocean response, may be investigated in future work.

CAM5, in common with most global climate models, diagnoses two categories of precipitation: the component that is diagnosed by the convection scheme is referred to as "convective precipitation"; while the component that is diagnosed by the large-scale stratiform cloud scheme is referred to as "large-scale precipitation". We refer to the sum of these two components as the "total precipitation".

Globally, the fire aerosols suppress total precipitation by $2\,\mathrm{mm/year}$ on average (Table 3). The convective and large-scale components are both suppressed by approximately $1\,\mathrm{mm/year}$ (Fig. 7a). Although black carbon and sulphur dioxide were found to make only a small contribution to the net RFP of the fire aerosols, these two species play a major role in the global



hydrological fast response to fire aerosols: black carbon and sulphur dioxide, rather than organic carbon, contribute to the suppression of convective precipitation; and black carbon exerts the strongest suppression of large-scale precipitation (Fig. 7a).

Regionally, the strongest suppression of total precipitation by black carbon occurs over the western tropical Pacific Ocean (Fig. S10b), with the caveat that statistical significance is absent. In contrast, organic carbon often plays a more important role in the hydrological fast response over many land regions (Fig. S10a, Fig. 7b–h). For example, our results suggest that a strong suppression of precipitation over Southern Hemisphere Africa is primarily driven by organic carbon emissions (Fig. 7f), in contrast to the conclusions of Hodnebrog et al. (2016) who suggest that black carbon also plays an important role.

For all seven regions shown in Fig. 7, the organic carbon emissions suppress both total precipitation and convective precipitation. The organic carbon emissions also suppress large-scale precipitation over five of the regions (Fig. 7d–h). The exceptions are the two Boreal regions, where large-scale precipitation is actually enhanced by the fire aerosols: over Boreal North America, the enhancement is driven by the organic carbon emissions (Fig. 7b); whereas over Boreal Asia, organic carbon, black carbon, and sulphur dioxide all appear to contribute (Fig. 7c).

Using inter-annually invariant fire emissions does not significantly affect the hydrological fast response of total precipitation either globally or over any of the regions, as shown by the lack of statistically significant "conventional"−"revised" differences in Table 3 and Fig. 8c. However, if an interactive ocean model were to be used, the "conventional"−"revised" RFP differences will likely impact SSTs which in turn will likely impact the hydrological slow response.

## 4 Discussion: how do the effects of inter-annually varying emissions differ from those of inter-annually invariant emissions?

Most global climate modelling studies use inter-annually invariant emissions of fire aerosols. In this study, the FMEAN simulation also followed this "conventional" approach of ignoring inter-annual variability. In contrast, the {Fyyyy} ensemble facilitates calculation of "revised" RFPs that take into account the inter-annual variability of the emissions. Consideration of "conventional"−"revised" RFP differences reveals the impact of the "conventional" approach of ignoring inter-annual variability of emissions.

Globally, the "conventional" approach causes net RFP strength to be overestimated by 23 % (Table 1, Fig. 5a). Over four of the fifteen regions (Fig. 1), the "conventional" approach also causes net RFP strength to be statistically significantly overestimated (Table 1). Over Australia and New Zealand, the overestimation is as large as 58 % (Fig. 5e). Over Boreal Asia, an overestimation of 43 % (Fig. 5c) corresponds to a "conventional"−"revised" RFP difference of $-1.9\,\mathrm{W\,m^{-2}}$ (Table 1).

When the net "conventional"−"revised" net RFP differences are calculated at the model output resolution of $1.9° \times 2.5°$, statistically significant negative differences occur across 5.2 % of the globe by area after controlling the False Discovery Rate (Benjamini and Hochberg, 1995; Wilks, 2016) (Fig. 6c). In contrast, statistically significant positive differences occur across only 0.5 % of the globe. There is no clear signal of positive "conventional"−"revised" differences. Hence, there is no clear evidence that the "conventional" approach of ignoring inter-annual variability ever leads to an underestimation of the net




RFP strength. Rather, across much of the globe, the "conventional" approach leads to a systematic overestimation of net RFP strength (Fig. 6c, Table 1), dominated by an overestimation of the cloud shortwave RFP strength (Table 2).

We hypothesise that the overestimation of cloud shortwave RFP strength occurs due to the non-linear influence of aerosols on clouds. At high concentration levels, the aerosol indirect effects may saturate. Therefore, compared to using inter-annually varying emissions, spreading out the aerosol emissions in time by averaging across different years will likely lead to a stronger indirect effect on average.

In order to test this hypothesis, we have produced scatterplots of cloud shortwave RFP versus surface organic carbon concentration for different months and locations within the Boreal Asia region (Fig. 9). Boreal Asia has been chosen because it is the region with the largest "conventional"−"revised" difference in cloud shortwave RFP (Table 2). The first location–month combination (Fig. 9a) has been selected because it is the location–month combination with the largest "conventional"−"revised" cloud shortwave RFP difference. The second location–month combination (Fig. 9b) is the combination with the next largest difference, with the additional criterion that the location is at least $10°$ (either longitude or latitude) from the previously selected location (Fig. 9a) so as to sample more widely geographically. Similarly, the third, fourth and fifth combinations (Fig. 9c–e) are those with the next largest differences at locations at least $10°$ from any of the previously selected locations. The method for selecting these five location–month combinations is shown in Fig. S12.

Looking at the scatterplots of cloud shortwave RFP versus surface organic carbon concentration (Fig. 9), the following observations can be made:

1. For all five location–month combinations, the surface organic carbon concentration is very similar between the "conventional" and "revised" approaches. On a larger scale, no significant "conventional"−"revised" difference in surface organic carbon concentration is found for either the globe or any of the fifteen regions (Table S4). This suggests that "conventional"−"revised" differences in cloud shortwave RFP are not driven by differences in aerosol lifetime. (For further discussion of how fire episodicity may impact the aerosol lifetime, see Clark et al., 2015).

2. For three of the location–month combinations (Fig. 9a–c), a non-linear relationship between surface organic carbon concentration and cloud shortwave RFP is evident. A logarithmic fit works well, with the cloud shortwave RFP scaling approximately linearly with the logarithm of the surface organic carbon concentration. The nature of the logarithmic relationship causes the mean of the {Fyyyy} ensemble to have a weaker cloud shortwave RFP than an individual {Fyyyy} ensemble member with a similar surface organic carbon concentration, an effect that is particularly evident in Fig. 9a,c. The remaining two location–month combinations (Fig. 9d,e) do not exhibit a clear relationship between surface organic carbon concentration and cloud shortwave RFP. For these two location–month combinations, the surface organic carbon concentration is not a good predictor of cloud shortwave RFP, possibly because surface organic carbon concentration may not always be a good proxy for the cloud condensation nuclei available to clouds.

3. For one of the location–month combinations (Fig. 9c), the cloud shortwave RFP for the FMEAN simulation is very similar to that of the {Fyyyy} ensemble member that has a similar surface organic carbon concentration. For this location–month combination, it appears that the non-linear relationship between surface organic carbon concentration



and cloud shortwave RFP can explain most of the large "conventional"−"revised" difference in cloud shortwave RFP. This supports our hypothesis that the "conventional" overestimation of cloud shortwave RFP strength occurs due to the non-linear influence of aerosols on clouds. The interpretation is less clear in Fig. 9a,b, where the FMEAN cloud shortwave RFP is stronger than would be expected for the given surface organic carbon concentration. Although the non-linear relationship between surface organic carbon concentration and cloud shortwave RFP can explain part of the "conventional"−"revised" difference in cloud shortwave RFP, it would appear that other factors also play a role.

It is worth noting that the results presented in Fig. 9 have two obvious limitations. First, they represent a relatively small sample of location–month combinations. Second, as pointed out above, surface organic carbon concentration may not always be a good proxy for the cloud condensation nuclei available to clouds.

Notwithstanding these two limitations, Fig. 9 (especially Fig. 9c) provides some evidence in support of our hypothesis that the overestimation of cloud shortwave RFP strength occurs due to the non-linear influence of aerosols on clouds. However, other factors might also play a role.

Despite the impact on net RFP, using inter-annually invariant emissions does not significantly impact the hydrological fast response, as shown by the absence of statistically significant "conventional"−"revised" differences in Table 3 and Fig 8c. The prescribed-SST simulations analysed here do not facilitate analysis of the hydrological slow response. However, if SST feedbacks were to be included, the "conventional"−"revised" RFP differences would likely impact surface temperature gradients. Changes in surface temperature gradients are known to impact precipitation patterns (Wang, 2015). Hence it is foreseeable that the "conventional" approach of ignoring inter-annual variability might influence the hydrological slow response.

## 5  Conclusions

In this study, we have investigated the radiative effects of inter-annually varying emissions of fire aerosols. Our prescribed-SST CAM5 simulation results suggest that fire aerosols exert a net radiative effect of $-1.0\,\mathrm{W\,m^{-2}}$ on the climate system. This net radiative effect is dominated by the cloud shortwave response to organic carbon emissions. Boreal regions are especially susceptible. For example, over Boreal North America, the net radiative effect of the fire aerosols is $-4.9\,\mathrm{W\,m^{-2}}$ on average and can be as strong as $-6.9\,\mathrm{W\,m^{-2}}$ in some years.

The "conventional" approach of ignoring the inter-annual variability of emissions can lead to a significant overestimation of the net radiative effect of the fire aerosols. Compared to our "revised" approach of using inter-annually varying emissions, the "conventional" approach of ignoring inter-annual variability can lead to a $+23\,\%$ overestimation of the global mean net radiative effect. Regionally, the "conventional" approach can lead to an even larger overestimation of the strength of the net radiative effect of the fire aerosols: $+32\,\%$ ($-1.6\,\mathrm{W\,m^{-2}}$) over Boreal North America, $+43\,\%$ ($-1.9\,\mathrm{W\,m^{-2}}$) over Boreal Asia, $+58\,\%$ ($-1.2\,\mathrm{W\,m^{-2}}$) over Australia and New Zealand, and $+49\,\%$ ($-0.8\,\mathrm{W\,m^{-2}}$) over the Western Maritime Continent.

There is evidence to suggest that the overestimation associated with the "conventional" approach arises due to the non-linear influence of aerosols on clouds. Compared to using inter-annually varying emissions (our "revised" approach), spreading out




the aerosol emissions in time by averaging across different years (the "conventional" approach) will likely lead to a stronger indirect effect on average.

Fire aerosols play an important role in the climate system, exerting large radiative effects regionally. In order to improve understanding of the climate system, we need to more accurately quantify these radiative effects, including critical characteristics

5  such as seasonality and inter-annual variability.

*Author contributions.* B.S.G. designed the experiment, performed the simulations, analysed the data, and wrote the manuscript. H.H.L. and C.W. contributed to the experimental design, analysis methodology, and redrafting of the manuscript.

*Data availability.* The CESM-CAM5 data analysed in this manuscript are available via Figshare: http://dx.doi.org/10.6084/m9.figshare.3497705.

*Acknowledgements.* This research was supported by the National Research Foundation (NRF), Prime Minister's Office, Singapore under its

10  Campus for Research Excellence and Technological Enterprise (CREATE) programme. The Center for Environmental Sensing and Modeling (CENSAM) is an interdisciplinary research group (IRG) of the Singapore-MIT Alliance for Research and Technology (SMART) centre. This research was also supported by the U.S. National Science Foundation (AGS-1339264), U.S. DOE (DE-FG02-94ER61937), and U.S. EPA (XA-83600001-1). The CESM project is supported by the U.S. National Science Foundation and the Office of Science (BER) of the U.S. Department of Energy. The GFED 4.0s dry matter emission data and emissions factors were downloaded from http://www.globalfiredata.org.



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





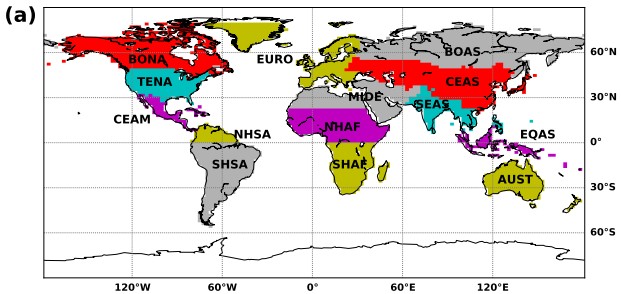

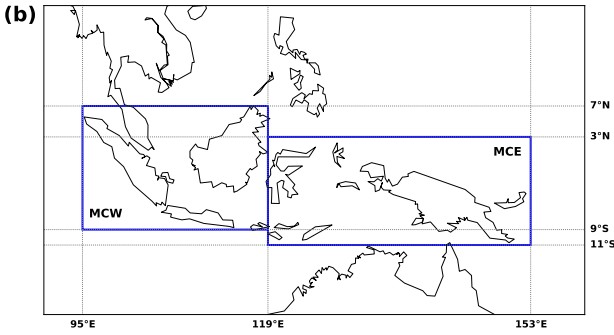

**Figure 1.** (a) The GFED4s basis-regions, re-gridded to the model resolution of approximately $1.9° \times 2.5°$ using nearest-neighbour interpolation: Boreal North America (BONA), Temperate North America (TENA), Central America (CEAM), Northern Hemisphere South America (NHSA), Southern Hemisphere South America (SHSA), Southern Hemisphere Africa (SHAF), Northern Hemisphere Africa (NHAF), the Middle East (MIDE), Europe (EURO), Boreal Asia (BOAS), Central Asia (CEAS), Southeast Asia (SEAS), Australia and New Zealand (AUST), and Equatorial Asia (EQAS). (b) The two regions used instead of EQAS in this study: the Western Maritime Continent (MCW), and the Eastern Maritime Continent (MCE). MCW and MCE both include ocean as well as land, unlike the land-only GFED4s basis-regions. In the other figures and tables, "Northern Hemisphere" is often abbreviated to "NH", while "Southern Hemisphere" is abbreviated to "SH".




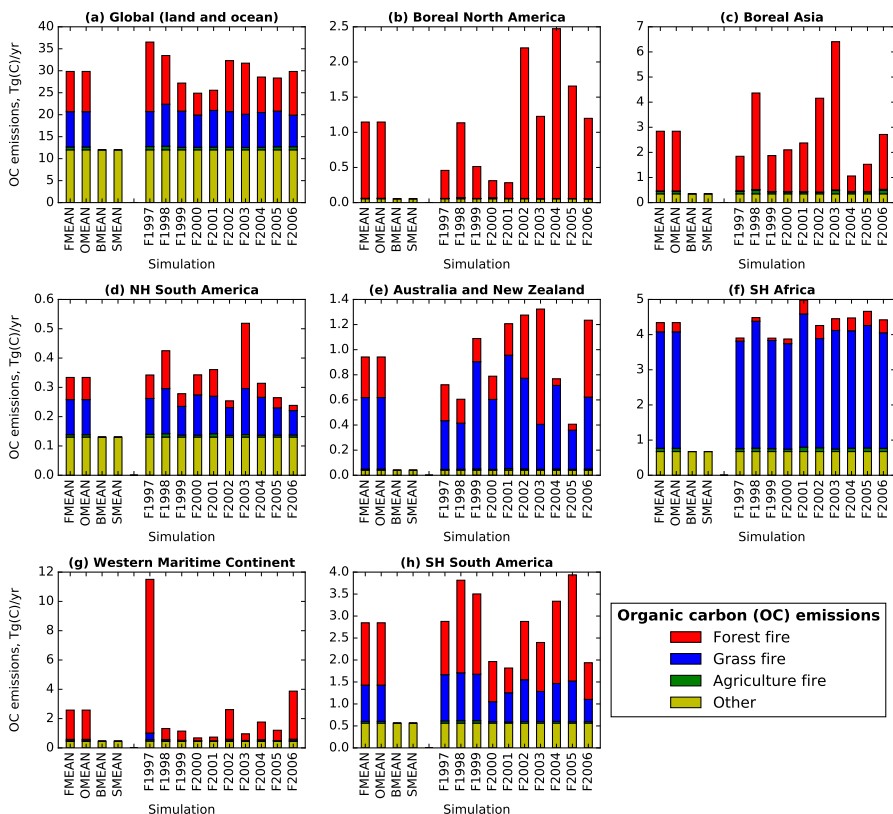

**Figure 2.** Organic carbon (OC) annual emissions for (a) the globe and (b)–(h) seven regions (see Fig. 1). The seven regions included here are those with the largest FMEAN−F0 net radiative flux perturbation differences (Table 1). "Other" refers to year-2000 emissions from non-fire sectors, such as domestic emissions and fossil fuel emissions from industry. Organic carbon emissions for another eight regions are shown in Fig. S1. Black carbon and sulphur dioxide emissions are shown in Figs. S2–S5.




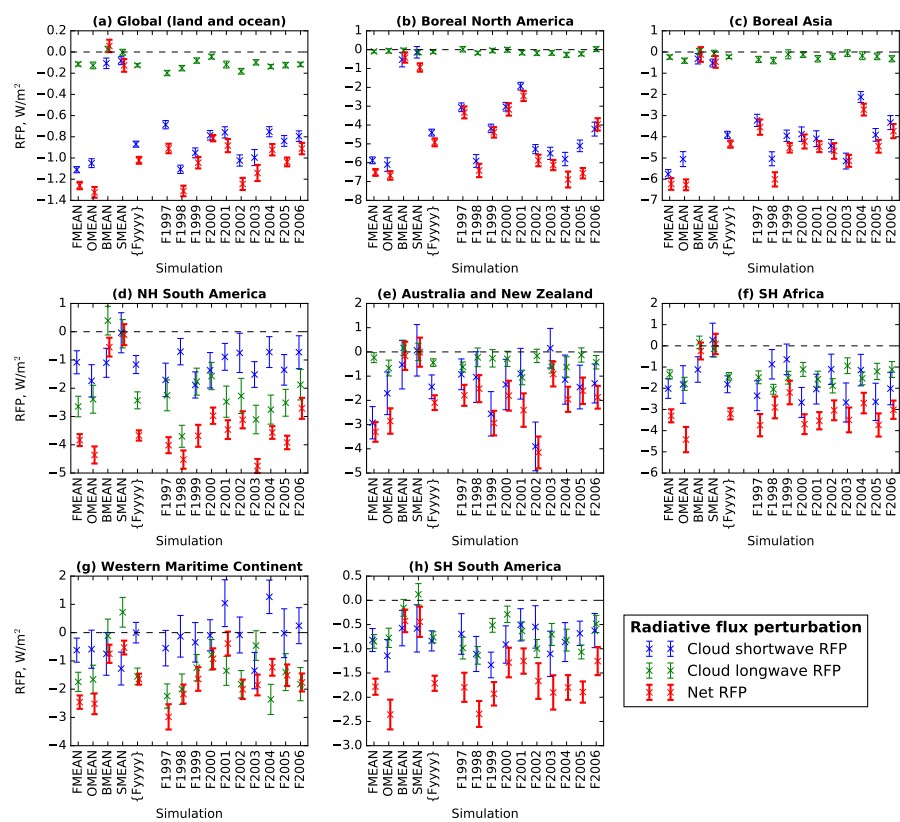

**Figure 3.** Cloud shortwave, cloud longwave, and net (all components, shortwave plus longwave) top-of-atmosphere radiative flux perturbations (RFPs) for the globe and seven regions. Results for another eight regions are shown in Fig. S6. The RFPs are relative to simulation F0. Error bars represent combined standard error – e.g. for the F1997 net RFP error bars, the combined standard error equals $\sqrt{\frac{s_{F0}^2}{N_{F0}} + \frac{s_{F1997}^2}{N_{F1997}}}$, where $s_{F0}$ and $s_{F1997}$ are the corrected sample standard deviations, and $N_{F0} = 42$ and $N_{F1997} = 12$ are the sample sizes.





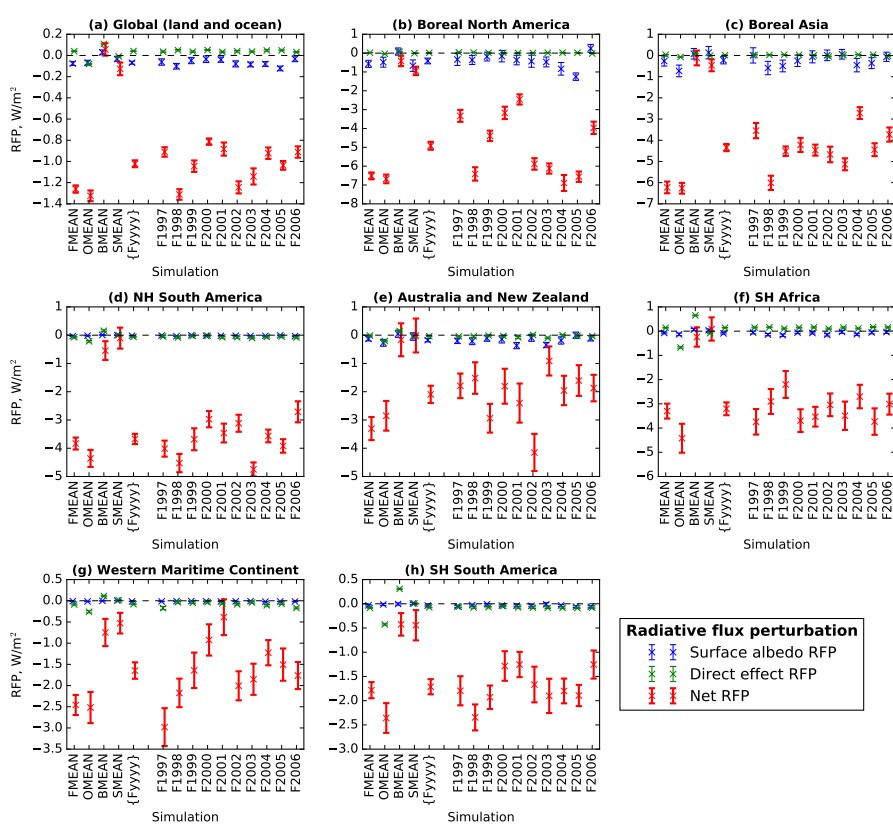

**Figure 4.** Surface albedo, aerosol direct effect, and net (all components, shortwave plus longwave) top-of-atmosphere radiative flux perturbations (RFPs) for the globe and seven regions. Results for another eight regions are shown in Fig. S7. The RFPs are relative to simulation F0. Error bars represent combined standard error (see Fig. 3 caption).





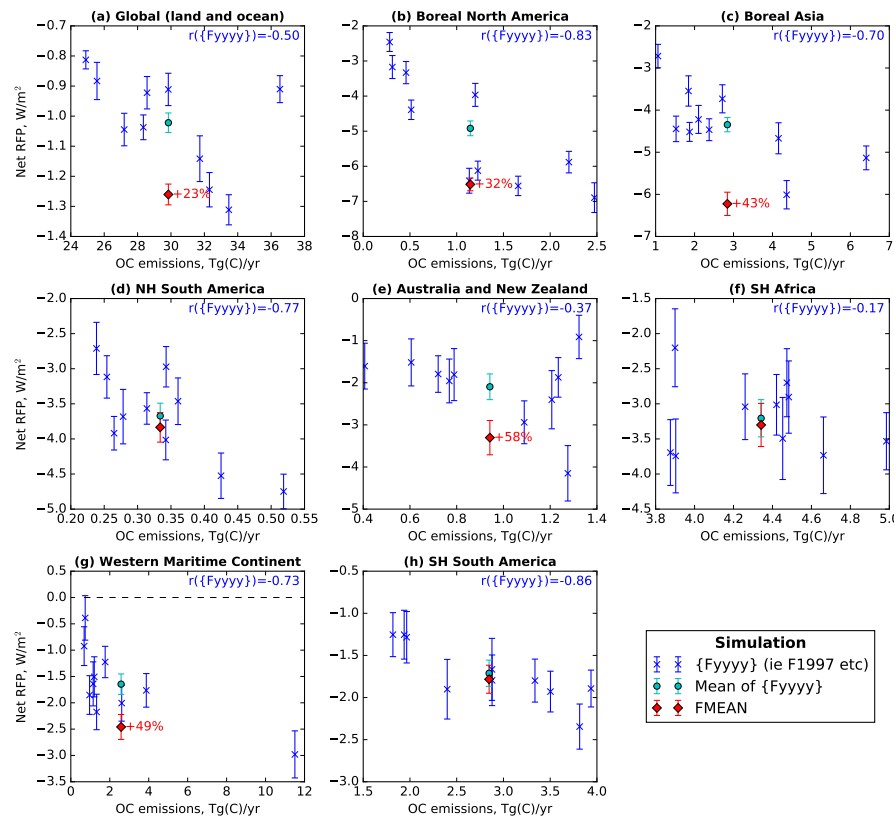

**Figure 5.** Net radiative flux perturbation (RFP) versus organic carbon (OC) emissions for the globe and seven different regions. The RFPs are relative to simulation F0. Each point represents the results for a different simulation or group of simulations (in the case of "Mean of {Fyyyy}"). Error bars represent combined standard error (see Fig. 3 caption). The Pearson's product-moment correlation coefficient ($r$), calculated from the ten-member {Fyyyy} ensemble (blue points), is shown in blue text at the top right of each panel. For regions where the FMEAN−{Fyyyy} difference is statistically significant at two-tailed $p < 0.05$ (tested using Welch's $t$-test, using annual mean data as the input), the FMEAN−{Fyyyy} percentage difference (relative to {Fyyyy}) is shown in red text.





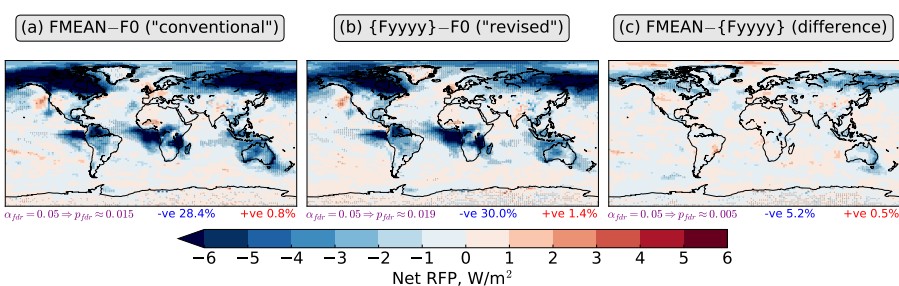

**Figure 6.** Net (shortwave plus longwave) top-of-atmosphere radiative flux perturbation (RFP) for (a) simulation FMEAN relative to simulation F0, (b) the {Fyyyy} ensemble relative to simulation F0, and (c) the FMEAN−{Fyyyy} difference. Stippling indicates differences that are statistically significant at a significance level of $\alpha_{fdr} = 0.05$ after controlling the False Discovery Rate (FDR; Benjamini and Hochberg, 1995; Wilks, 2016). The two-tailed $p$-values are generated by Welch's $t$-test, using annual mean data as the input. The approximate $p$-value threshold, $p_{fdr}$, is written in purple underneath each map. The percentage of the globe (area-weighted) over which negative ("-ve") statistically significant differences occur is written in blue text underneath each map, while the percentage of the globe over which positive ("+ve") statistically significant differences occur is written in red text.





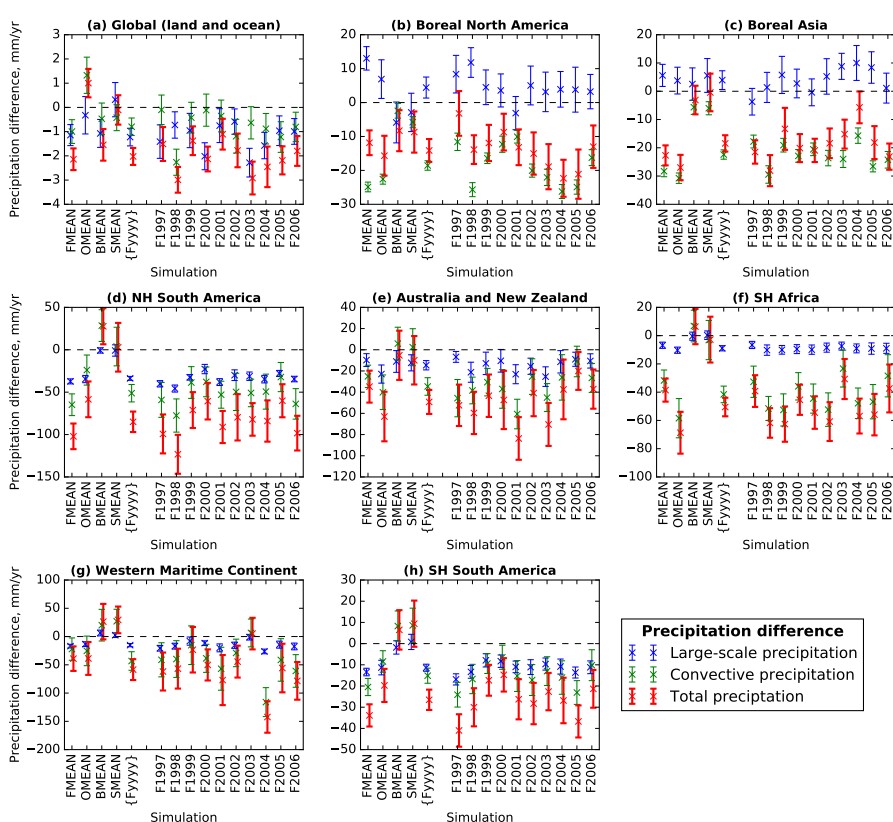

**Figure 7.** Large-scale, convective, and total (large-scale plus convective) annual precipitation differences, relative to simulation F0, for the globe and seven regions. Error bars represent combined standard error. Results for another eight regions are shown in Fig. S8.





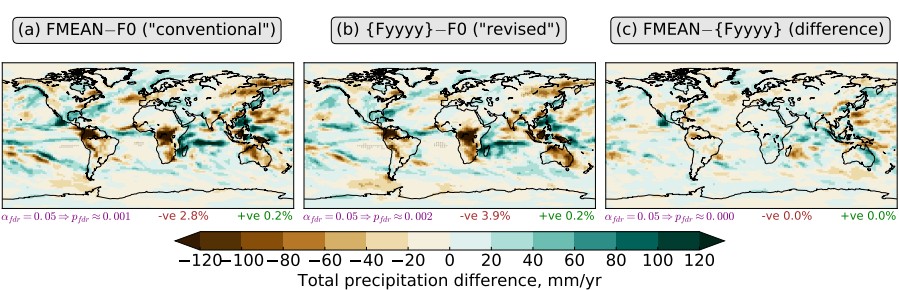

**Figure 8.** Total (large-scale plus convection) annual precipitation differences for (a) simulation FMEAN relative to simulation F0, (b) the {Fyyyy} ensemble relative to simulation F0, and (c) the FMEAN−{Fyyyy} difference. Stippling indicates differences that are statistically significant at a significance level of $\alpha_{fdr} = 0.05$ after controlling the False Discovery Rate (FDR; Benjamini and Hochberg, 1995; Wilks, 2016). The two-tailed $p$-values are generated by Welch's $t$-test, using annual mean data as the input. The approximate $p$-value threshold, $p_{fdr}$, is written in purple underneath each map. The percentage of the globe (area-weighted) over which negative ("-ve") statistically significant differences occur is written in brown text underneath each map, while the percentage of the globe over which positive ("+ve") statistically significant differences occur is written in green text.





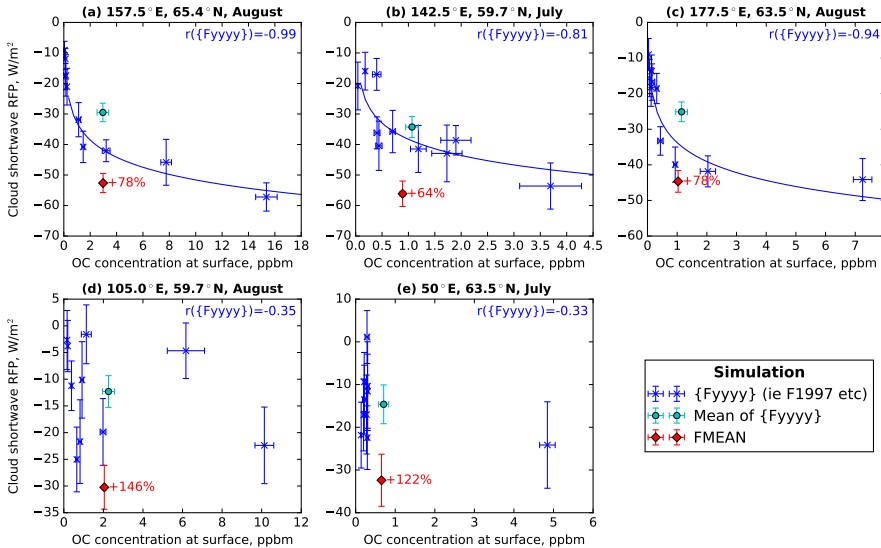

**Figure 9.** Cloud shortwave top-of-atmosphere radiative flux perturbation (RFP) versus surface organic carbon (OC) concentration for different months and locations within the Boreal Asia region. The five location–month combinations have been selected according to the procedure described in Fig. S12, based on the following three criteria: (1) within the Boreal Asia region, select the location–month combinations with (2) the largest FMEAN−{Fyyyy} cloud shortwave RFP differences, (3) at least $10°$ (either longitude or latitude) from one another. The RFPs are relative to simulation F0. The organic carbon concentrations are given in parts-per-billion by mass (1 ppbm $= 10^{-9}$ kg/kg). Each point represents the results for a different simulation or group of simulations (in the case of "Mean of {Fyyyy}"). Error bars represent combined standard error (see Fig. 3 caption). In (a)–(c), the blue line shows the results of a linear regression fit to $y = a + b\ln x$, calculated from the ten-member {Fyyyy} ensemble (blue points). In both (d) and (e), the regression fit was found to be statistically insignificant at the $\alpha = 0.05$ significance level and is therefore not shown. The corresponding Pearson's product-moment correlation coefficient ($r$) for $y = a + b\ln x$, calculated from the ten-member {Fyyyy} ensemble, is shown in blue text at the top right of each panel. For regions where the FMEAN−{Fyyyy} difference is statistically significant at two-tailed $p < 0.05$ (tested using Welch's $t$-test, using annual mean data as the input), the FMEAN−{Fyyyy} percentage difference (relative to {Fyyyy}) is shown in red text.





**Table 1.** Net (shortwave plus longwave) top-of-atmosphere radiative flux perturbation (RFP) differences for different simulation combinations and regions (see Fig. 1). Combined standard errors have been calculated using corrected sample standard deviations – e.g. the standard error for the global {Fyyyy}−F0 difference was calculated using $\sqrt{\frac{s^2_{\{Fyyyy\}}}{N_{\{Fyyyy\}}} + \frac{s^2_{F0}}{N_{F0}}}$, where $s_{\{Fyyyy\}}$ and $s_{F0}$ are the corrected sample standard deviations, and $N_{\{Fyyyy\}} = 120$ and $N_{F0} = 42$ are the sample sizes. Significance has been tested using Welch's $t$-test: '*' indicates differences that are statistically significant at two-tailed $p < 0.05$; and '**' indicates differences that are statistically significant at two-tailed $p < 0.01$. The regions have been ranked according the FMEAN−F0 "conventional" net RFP strength. A horizontal line is used to indicate the seven regions that receive more focus in the Results section.

| Region | Net radiative flux perturbation (RFP), W m$^{-2}$ | | |
| --- | --- | --- | --- |
| | FMEAN−F0 "conventional" | {Fyyyy}−F0 "revised" | FMEAN−{Fyyyy} difference |
| Global (land and ocean) | $-1.26 \pm 0.03$** | $-1.02 \pm 0.03$** | $-0.24 \pm 0.03$** |
| Boreal North America | $-6.52 \pm 0.18$** | $-4.92 \pm 0.21$** | $-1.60 \pm 0.21$** |
| Boreal Asia | $-6.23 \pm 0.28$** | $-4.35 \pm 0.17$** | $-1.88 \pm 0.27$** |
| NH South America | $-3.83 \pm 0.21$** | $-3.67 \pm 0.18$** | $-0.16 \pm 0.18$ |
| Australia and New Zealand | $-3.30 \pm 0.41$** | $-2.10 \pm 0.31$** | $-1.21 \pm 0.36$** |
| SH Africa | $-3.30 \pm 0.31$** | $-3.21 \pm 0.27$** | $-0.10 \pm 0.26$ |
| Western Maritime Continent | $-2.46 \pm 0.24$** | $-1.65 \pm 0.19$** | $-0.81 \pm 0.22$** |
| SH South America | $-1.78 \pm 0.17$** | $-1.71 \pm 0.16$** | $-0.07 \pm 0.13$ |
| Central America | $-1.60 \pm 0.29$** | $-1.14 \pm 0.26$** | $-0.46 \pm 0.22$* |
| Eastern Maritime Continent | $-1.45 \pm 0.20$** | $-1.22 \pm 0.18$** | $-0.24 \pm 0.17$ |
| Temperate North America | $-0.98 \pm 0.32$** | $-1.25 \pm 0.26$** | $+0.27 \pm 0.28$ |
| Central Asia | $-0.92 \pm 0.28$** | $-0.73 \pm 0.22$** | $-0.19 \pm 0.22$ |
| NH Africa | $-0.79 \pm 0.17$** | $-0.73 \pm 0.13$** | $-0.06 \pm 0.16$ |
| Southeast Asia | $-0.61 \pm 0.23$** | $-0.23 \pm 0.20$ | $-0.39 \pm 0.19$* |
| Europe | $-0.27 \pm 0.33$ | $-0.14 \pm 0.24$ | $-0.13 \pm 0.28$ |
| Middle East | $-0.14 \pm 0.12$ | $-0.08 \pm 0.10$ | $-0.07 \pm 0.10$ |



**Table 2.** Cloud shortwave top-of-atmosphere radiative flux perturbation (RFP) differences for different simulation combinations and regions. Combined standard errors have been calculated. Significance has been tested using Welch's $t$-test: '*' indicates differences that are statistically significant at two-tailed $p < 0.05$; and '**' indicates differences that are statistically significant at two-tailed $p < 0.01$.

| Region | Cloud shortwave radiative flux perturbation (RFP), $\mathrm{W\,m^{-2}}$ | | |
| | FMEAN$-$F0 | {Fyyyy}$-$F0 | FMEAN$-${Fyyyy} |
| | "conventional" | "revised" | difference |
| --- | --- | --- | --- |
| Global (land and ocean) | $-1.11 \pm 0.03$** | $-0.87 \pm 0.03$** | $-0.24 \pm 0.03$** |
| Boreal North America | $-5.88 \pm 0.18$** | $-4.40 \pm 0.18$** | $-1.47 \pm 0.20$** |
| Boreal Asia | $-5.76 \pm 0.21$** | $-3.91 \pm 0.17$** | $-1.85 \pm 0.21$** |
| NH South America | $-1.08 \pm 0.41$** | $-1.16 \pm 0.32$** | $+0.08 \pm 0.33$ |
| Australia and New Zealand | $-2.92 \pm 0.67$** | $-1.44 \pm 0.49$** | $-1.48 \pm 0.59$* |
| SH Africa | $-2.01 \pm 0.46$** | $-1.81 \pm 0.40$** | $-0.20 \pm 0.39$ |
| Western Maritime Continent | $-0.62 \pm 0.43$ | $-0.00 \pm 0.36$ | $-0.62 \pm 0.36$ |
| SH South America | $-0.82 \pm 0.24$** | $-0.84 \pm 0.21$** | $+0.02 \pm 0.18$ |
| Central America | $-1.75 \pm 0.47$** | $-1.08 \pm 0.41$* | $-0.68 \pm 0.37$ |
| Eastern Maritime Continent | $-0.62 \pm 0.37$ | $-0.72 \pm 0.33$* | $+0.09 \pm 0.30$ |
| Temperate North America | $-1.05 \pm 0.37$** | $-1.30 \pm 0.31$** | $+0.26 \pm 0.31$ |
| Central Asia | $-0.30 \pm 0.28$ | $-0.22 \pm 0.23$ | $-0.08 \pm 0.23$ |
| NH Africa | $-0.28 \pm 0.35$ | $-0.17 \pm 0.28$ | $-0.11 \pm 0.28$ |
| Southeast Asia | $-0.20 \pm 0.44$ | $+0.41 \pm 0.37$ | $-0.61 \pm 0.37$ |
| Europe | $+0.48 \pm 0.41$ | $+0.10 \pm 0.30$ | $+0.38 \pm 0.36$ |
| Middle East | $-0.37 \pm 0.30$ | $-0.12 \pm 0.21$ | $-0.25 \pm 0.25$ |





**Table 3.** Total (large-scale plus convective) annual precipitation differences for different simulation combinations and regions. Combined standard errors have been calculated. Significance has been tested using Welch's $t$-test: '*' indicates differences that are statistically significant at two-tailed $p < 0.05$; and '**' indicates differences that are statistically significant at two-tailed $p < 0.01$.

| Region | Total precipitation difference, mm yr$^{-1}$ | | |
| --- | --- | --- | --- |
| | FMEAN−F0 | {Fyyyy}−F0 | FMEAN−{Fyyyy} |
| | "conventional" | "revised" | difference |
| Global (land and ocean) | $-2.1 \pm 0.4$** | $-2.0 \pm 0.4$** | $-0.1 \pm 0.4$ |
| Boreal North America | $-11.8 \pm 3.6$** | $-14.1 \pm 3.4$** | $+2.3 \pm 2.7$ |
| Boreal Asia | $-22.7 \pm 3.5$** | $-18.5 \pm 2.9$** | $-4.2 \pm 3.0$ |
| NH South America | $-102.1 \pm 15.1$** | $-85.0 \pm 12.1$** | $-17.1 \pm 12.5$ |
| Australia and New Zealand | $-34.8 \pm 15.1$* | $-49.2 \pm 11.3$** | $+14.4 \pm 13.2$ |
| SH Africa | $-38.5 \pm 8.2$** | $-50.5 \pm 6.5$** | $+12.0 \pm 7.2$ |
| Western Maritime Continent | $-39.1 \pm 21.7$ | $-58.5 \pm 18.4$** | $+19.4 \pm 18.2$ |
| SH South America | $-33.9 \pm 5.2$** | $-26.5 \pm 4.8$** | $-7.4 \pm 4.1$ |
| Central America | $+19.5 \pm 10.6$ | $+12.0 \pm 9.1$ | $+7.5 \pm 8.4$ |
| Eastern Maritime Continent | $-6.5 \pm 16.4$ | $-2.6 \pm 14.6$ | $-3.9 \pm 13.0$ |
| Temperate North America | $-2.5 \pm 8.7$ | $+7.7 \pm 7.7$ | $-10.1 \pm 6.5$ |
| Central Asia | $-6.3 \pm 5.1$ | $-2.9 \pm 4.2$ | $-3.4 \pm 4.1$ |
| NH Africa | $-31.9 \pm 10.0$** | $-33.1 \pm 7.9$** | $+1.2 \pm 8.0$ |
| Southeast Asia | $-12.5 \pm 14.0$ | $-18.3 \pm 10.9$ | $+5.8 \pm 11.9$ |
| Europe | $-6.5 \pm 5.7$ | $-5.0 \pm 4.4$ | $-1.5 \pm 4.7$ |
| Middle East | $+8.4 \pm 5.5$ | $+3.0 \pm 4.0$ | $+5.3 \pm 4.6$ |