# Peer review of "Radiative effects of inter-annually varying versus inter-annually invariant aerosol emissions from fires"

_Atmospheric Chemistry and Physics, 2016_

## Referee Comment (RC1) · D. S. Ward (Referee) · 10 Aug 2016

Grandey et al. compute the global radiative effects of fire aerosol emissions in a series of simulations with CAM5. In this experiment they compare the radiative effects that result when a monthly climatology is used to drive the emissions, to the radiative effects that result when emissions include interannual variability. The magnitude of the forcing is greater when the monthly climatology is used, suggesting a saturation of the first indirect aerosol effect, and shedding light on previous estimates of fire aerosol radiative forcings which typically use the climatology approach. There is no similarly large difference in precipitation between the two emissions schemes. Results are broken down by aerosol species and by region which highlights the greater interannual variability of

fires in boreal regions compared to the tropics and subtropics.

The design of the experiment is thoughtful and well suited to answer the questions posed in the introduction. The main result, a 20% decrease in RF magnitude when using the interannually varying fire emissions in this model setup, has implications beyond fire modeling and for Earth System modeling since many major ESM projects use decadal mean fire emissions. I have some comments, mostly minor, and suggestions for discussing the saturation effect differently.

General comments:

1. The argument in the Discussion section (Pg 11 Lines 3-6) for saturation of the aerosol radiative effect on clouds for the high fire emission years and locations (as opposed to the steady mean emissions) seems right, but the test of this hypothesis, illustrated with Figure 9, is not as convincing. Figure 9 shows how the sublinear response of the indirect effect to increases in aerosols is evident for a few grid points in a region, but does not give reason to think that the emissions vs. RFP relationship at these three grid points should apply generally. Especially give that two of the five points selected do not show the same relationship. And since the five points in the figure were already chosen specially because they had the largest FMEAN-F{yyyy} difference (among the other criteria), the figure does not show that the saturation effect matters here necessarily.

What if you were to plot timeseries (maybe monthly?) of Boreal Asia region average OC emissions, CCN (I think there is a standard CCN 0.2%SS variable output in CAM5?) and RFP, or some metric of forcing efficiency, with each timeseries standardized so they could be plotted together. Just one F{yyyy} could be plotted with FMEAN I think. This could show whether peaks and valleys in the interannual varying OC/CCN timeseries are matched by similar increases and decreases in RFP or if this saturates at the high extremes as would be expected. I am not sure if this effect would be clearly visible on a regional average but this might be something to try.

I also suggest including a standard reference for the saturation effect (maybe at Pg 11, Line 4) such as Boucher and Pham (2002). And since the first indirect effect is not isolated from all aerosol/cloud effects in this study, i.e. lifetime effect, semi-direct effect, it would be helpful to include discussion of how the additional aerosol effects factor in to estimates of the saturation effect. Or how they cloud the conclusions that can be drawn.

2. Related to this, I think it would be helpful to include more details about how aerosols interact with the radiative budget in this model setup, probably in the Methods section 2.1. For example, mentioning the different effects that are included in the RFP such as the direct effect, semi-direct effect, etc., explaining how aerosols can perturb the radiative flux in this model setup. This is also important to explain for the hydrological response which is discussed in Section 3.2 and shown in Figure 8. Aerosol impacts on both stratiform and convective clouds are included even though aerosols in the model do not directly interact with the convective scheme (Section 2.1). So here is another instance where explaining how the aerosols indirectly impact convection in the methods would be helpful.

Minor comments:

Pg 5, Lines 15-17: There are not many estimates of fire RF but you could include Chuang et al. (2002), -1.16 Wm-2 (first indirect effect only), and Ward et al. (2012), -1.38 Wm-2 (also CAM5), as additional points of comparison.

Pg 6, Line 9: The foundation for the rest of the results section is developed very nicely in this section. For this line, please include how you define statistical significance here.

Pg 6, Lines 16-17: It is mentioned here and shown in Figure 5 that fire aerosol impacts on forcing extend over ocean but it seems like only land area is considered in the regional estimates? In Ward et al. (2012) we also found large forcings (< -5 Wm-2) over ocean for fire aerosols, in fact, globally the largest indirect aerosol forcings were for the marine stratocumulus decks off the coasts of western Africa and South America.

If forcings over ocean are not considered here, can you discuss how the answers might be different if these areas were included?

Pg 7, Lines 26-27: I wonder if you can speculate why the BC emissions have a higher cloud sw forcing efficiency in these regions, and why the BC sw cloud forcing does not have a consistent sign in general when looking at all global regions. Perhaps this is a result of semi-direct effects (e.g. Sakaeda et al., 2011; Koch and Del Genio, 2010).

Pg 8, Lines 4-6: This summary point for the South American region is clear and nicely explained. I think it is important to note again here that aerosols are only microphysically active for stratiform clouds. This could be especially important in the tropical regions.

Pg 9, Line 1: Why is EQAS divided into two sub-regions for this study?

Pg 10, Lines 9-12: This sentence left me wondering why this is the case – that is that the fire aerosols lead to a precipitation increase. I am not sure it is feasible to tease out the reasons for this given the scope of this study, but it might be worth looking at what other changes are occurring such as changes in cloud fraction, cloud depths, CDNC, etc. that might provide some insight. Or maybe this has to do with differences in snow cover due to fire aerosols.

Pg 10, Lines 15-16: This is a good point, and might be a good place to note that Tosca et al. (2013) and Clark et al. (2015) do show a dynamical response in precipitation to fire aerosols using a slab ocean, although with non-interannually varying emissions. Another place these could be cited would be on/near Pg 9, Line 24.

Pg 13, Line 8: It is great that the data from this study are made available and so easily accessible.

References:

Boucher, O. and M. Pham, History of sulfate aerosol radiative forcings, Geophys. Res. Lett., 29(9), doi:10.1029/2001GL014048, 2002.

[Figure]

Chuang, C. C., Penner, J. E., Prospero, J. M., Grant, K. E., Rau, G. H., Kawamoto, K.: Cloud susceptibility and the first aerosol indirect forcing: Sensitivity to black carbon and aerosol concentrations, J. Geophys. Res., 107, 4564, doi:10.1029/2000JD000215, 2002.

Clark, S. K., Ward, D. S., and Mahowald, N. M.: The sensitivity of global climate to the episodicity of fire aerosol emissions, Journal of Geophysical Research: Atmospheres, 120, 11,589–11,607, doi:10.1002/2015JD024068, 2015.

Koch, D. and Del Genio, A. D.: Black carbon semi-direct effects on cloud cover: review and synthesis, Atmos. Chem. Phys., 10, 7685–7696, doi:10.5194/acp-10-7685-2010, 2010.

Sakaeda, N., Wood, R., and Rasch, P. J.: Direct and semidirect aerosol effects of southern African biomass burning aerosol, J. Geophys. Res.-Atmos., 116, D12205, doi:10.1029/2010JD015540, 2011.

Tosca, M., J. Randerson, and C. Zender (2013), Global impact of contemporary smoke aerosols from landscape fires on climate and the hadley circulation, Atmos. Chem. Phys., 12, 28,069–28,108.

Ward, D., S. Kloster, N. Mahowald, B. Rogers, J. Randerson, and P. Hess (2012), The changing radiative forcing of fires: Global model estimates for past, present and future, Atmos. Chem. Phys., 12, 10,857–10,886.

---

## Referee Comment (RC2) · A. Veira (Referee) · 23 Aug 2016

Aerosol emissions from open-burning fires show a distinct inter-annual variability. This manuscript by Grandey et al. quantifies the impact of the inter-annual fire emission variability on global and regional radiative effects. The authors present results from simulations with the Community Atmosphere Model version 5 (CAM5) which compare top-of-atmosphere radiative flux pertubations for inter-annually varying and inter-annually invariant emission inventories. The application of inter-annually varying fire emissions reduces the mean net radiative aerosol effect by about 23% globally and up to 58% regionally. The simulated global changes in precipitation, however, are not significant.

The scientific topics addressed by this manuscript are within the scope of ACP and

the study design is adequate for investigation of the given research questions. Overall, the paper is well structured and well written. The results are presented in a clear and understandable way, supported by neat figures and tables. The detailed discussion of the individual regional impacts appears slightly long-winded in some parts, but it provides substantial information for the overall conclusions of the study and might as well be beneficial for subsequent future studies. The reference list is comprehensive and largely reasonable. Due to the large differences in the overall fire aerosol radiative forcing between CAM5 and other global aerosol models, slightly more detailed descriptions and discussions of the used emission inventory uncertainties, the applied microphysical aerosol properties and the cloud shortwave radiative flux pertubations in the model, are desirable. In addition, I have some minor comments and suggestions.

I recommend publication in ACP if the authors sufficiently address the following comments in their revised manuscript.

General Comments:

1. The fire emission data set GFED applied in this study is a well-established and frequently used fire emission data set and there are no objections to use it for the purpose of this study. However, fire emission estimates based on other retrieval methods (e.g. the Global Fire Assimilation System GFAS, Kaiser et al., 2012) provide emission estimates which show substantial regional emission flux differences compared to GFED (e.g. Kaiser et al., 2012 and Zhang et al., 2014). Therefore I suggest at least to mention the emission estimate uncertainties in more detail in Section 2.2 and in the discussion of the results.

2. Globally, the radiative flux perturbation (RFP) of more than 1 Wm-2 of both, the 'conventional' and the 'revised' fire emission approaches presented in Section 3.1.1 are surprisingly large compared to most other studies which provide top-of-atmosphere radiative forcings of fire emissions which range between -0.3 Wm-2 and +0.1 Wm-2 (for a list of references see Veira et al., 2015, Section 5, page 7188). Can you provide a

more detailed explanation, why this is the case? The comparatively large values of the anthropogenic / total RFP in CAM5.1 provided by Shindell et al., 2013 (cited on P3, L16 and P5, L18 in this manuscript) do not necessarily explain the very large radiative effect of the fire aerosol emissions in particular. Might the black carbon (BC) and organic carbon (OC) atmospheric lifetime largely differ in the other models? Which refractive indices for BC and OC are used in CAM5 (for a comparison to ECHAM-HAM2, see Zhang et al., 2012)? What specific implementation in the cloud microphysics is included in CAM5, but not in the other models?

3. Cloud shortwave RFP is identified as major driver of the total net RFP differences between the simulations with inter-annually invariant and inter-annually varying fire emissions for many regions in the results section. Can you provide any numbers on high and low cloud cover changes in these regions? Do you see distinct differences in vertical cloud profiles? Some numbers would be helpful to further explain the non-linear saturation effects which are nicely presented in the discussion section.

Specific Comments:

Abstract:

P1, L14-15: '[...], we need to more accurately quantify the effects of aerosols [...]'. This sub-clause is a very general statement and does not provide specific outcomes of this study. Therefore the last sentence of the abstract can be shortened by pointing directly to the specific importance of the inter-annual variability of fire aerosols quantified in this study.

Introduction:

P1, L21-22: The references referring to peat fires in Indonesia are slightly misleading in this context. Although the sulfur content of the biomass varies substantially, all fire categories named in line 16-17 include emissions of sulfur dioxide as the emission factor inventories by Andreae and Merlet, 2001 and Akagi et al., 2011 demonstrate.

P1, L22-23: Does the statement 'These aerosols have a negative impact on [. . .]' refer to sulfur dioxide emissions from fire? It should be clarified that Lelieveld et al., 2015 quantified the overall impact of biomass burning aerosols (including black carbon and organic carbon) on human health, not the human health impact of sulfur emissions from fires in particular.

P2, L1: 'Aerosols scatter or absorb incoming sunlight'. The conjunction 'or' is misleading as both processes, scattering and absorption, apply to most atmospheric aerosol species.

P2, L7-8: '[. . .] many other aerosol effects on clouds have also been proposed.' Please provide examples of these 'other' effects.

P2, L11 and L14: In the reference list, there are two references given for Wang et al., 2009. Therefore it is unclear, which of the two papers is referred to in line 11 and line 14, respectively.

Methods:

P4, L14-18: In this paragraph, it should be explicitly stated that the black carbon and organic carbon fire emission estimates from GFED applied in this study represent an important source of uncertainty (see general comment 1).

Results:

P5, L26: Can you provide numbers for the increase in snow cover? It would be interesting to see the quantitative relationship between snow cover increase and surface albedo change.

P9, L19: Comparing Figure S1 to Table 1, it is interesting to see that a statistically significant overestimation of the net RFP strength is found for Central America and Southeast Asia. In contrast, the Eastern Maritime Continent and Temperate North America show temporal OC emission patterns of similiar magnitude like those of Central America and Southeast Asia, respectively (Figure S1), but nevertheless no statistically

significant overestimation of the net RFP is simulated for the conventional approach in these regions. Might these findings indicate that the influence of the meteorology varies significantly in these regions or is this due to the saturation effects?

P10, L1-2: Why does black carbon suppress large-scale precipitation in such a way? If available, an analysis of vertical temperature profiles / atmospheric stability changes might be helpful for a better understanding although the discussion of these relations does of course not represent the focus of this study.

Discussion:

P11, L1: 'Rather, across much of the globe, the "conventional" approach leads to a systematic overestimation of net RFP [...]'. Table 1 and Fig. 6c show that the statistically significant overestimation is largely limited to some specific regions. Therefore the use of the statement 'much of the globe' is inappropriate in this context.

P12, L12: What 'other factors' do you think of? Might interaction with other, non-fire related aerosol species play a role?

Conclusions:

P12, L28-30: This paragraph represents a nice description of those regions, where ignoring the inter-annual variability leads to an overestimation of the net radiative effect of fire aerosols. However, as shown in Table 1, there are many other regions, where no significant differences between the two scenarios are found and this should also be stated here.

P13, L3-5: At this point I suggest to provide some implications and ideas for subsequent future studies, which could be initiated by the results and conclusions presented in this paper, e.g. the quantification of daily vs. monthly fire emissions or a deeper analysis of the cloud micro-physical processes which are responsible for the found saturation effects.

All figures presented in the manuscript represent elaborated and easily readable visualizations of high quality with detailed figure captions. Figure 6 and Figure 8 are neat and self-explanatory figures, but in the final manuscript the figure size should be increased in order to improve the readability of the stippling.

References:

Akagi, S. K., Yokelson, R. J., Wiedinmyer, C., Alvarado, M. J., Reid, J. S., Karl, T., Crounse, J. D., and Wennberg, P. O.: Emission factors for open and domestic biomass burning for use in atmospheric models, Atmos. Chem. Phys., 11, 4039-4072, doi:10.5194/acp-11-4039-2011, 2011.

Andreae, M. O., and P. Merlet (2001), Emission of trace gases and aerosols from biomass burning, Global Biogeochem. Cycles, 15(4), 955–966, doi:10.1029/2000GB001382.

Kaiser, J. W., Heil, A., Andreae, M. O., Benedetti, A., Chubarova, N., Jones, L., Morcrette, J.-J., Razinger, M., Schultz, M. G., Suttie, M., and van der Werf, G. R.: Biomass burning emissions estimated with a global fire assimilation system based on observed fire radiative power, Biogeosciences, 9, 527-554, doi:10.5194/bg-9-527-2012, 2012.

Zhang, K., O'Donnell, D., Kazil, J., Stier, P., Kinne, S., Lohmann, U., Ferrachat, S., Croft, B., Quaas, J., Wan, H., Rast, S., and Feichter, J.: The global aerosol-climate model ECHAM-HAM, version 2: sensitivity to improvements in process representations, Atmos. Chem. Phys., 12, 8911-8949, doi:10.5194/acp-12-8911-2012, 2012.

Zhang, F.,Wang, J., Ichoku, C., Hyer, E. J., Yang, Z., Ge, C., Su, S., Zhang, X., Kondragunta, S., Kaiser, J. W., Wiedinmyer, C., and da Silva, A.: Sensitivity of mesoscale modeling of smoke direct radiative effect to the emission inventory: a case study in northern sub-Saharan African region, Environ. Res. Lett., 9, 075002, doi:10.1088/1748-9326/9/7/075002, 2014.

---

## Author Comment (AC1) · 12 Oct 2016

**Response to referee comments on *Radiative effects of inter-annually varying versus inter-annually invariant aerosol emissions from fires**

B. S. Grandey, H.-H. Lee, and C. Wang

October 12, 2016

We thank the two referees for taking the time to evaluate the manuscript and to provide comments. Please find the referee comments (shown in blue) and our responses (shown in black) below. Quotes from the revised manuscript are shown in red.

**1 Response to Referee #1 (D. S. Ward)**

Grandey et al. compute the global radiative effects of fire aerosol emissions in a series of simulations with CAM5. In this experiment they compare the radiative effects that result when a monthly climatology is used to drive the emissions, to the radiative effects that result when emissions include interannual variability. The magnitude of the forcing is greater when the monthly climatology is used, suggesting a saturation of the first indirect aerosol effect, and shedding light on previous estimates of fire aerosol radiative forcings which typically use the climatology approach. There is no similarly large difference in precipitation between the two emissions schemes. Results are broken down by aerosol species and by region which highlights the greater interannual variability of fires in boreal regions compared to the tropics and subtropics.

The design of the experiment is thoughtful and well suited to answer the questions posed in the introduction. The main result, a 20% decrease in RF magnitude when using the interannually varying fire emissions in this model setup, has implications beyond fire modeling and for Earth System modeling since many major ESM projects use decadal mean fire emissions. I have some comments, mostly minor, and suggestions for discussing the saturation effect differently.

Thank you for taking the time to thoughtfully read the manuscript and for providing helpful suggestions. We respond to your comments below.

Incidentally, we think your use of "monthly climatology" is a clear way to describe our FMEAN simulation. Hence, we have added the following sentence to the description of FMEAN in the Method section: "In other words, the FMEAN emissions consist of a monthly climatology." We have also incorporated "monthly climatology" elsewhere, e.g. in the Abstract ("Conventionally, many climate modelling studies have used an inter-annually invariant monthly climatology of emissions of fire aerosols.") and at the start of the Results section ("In most global climate modelling studies, a monthly climatology of aerosol emissions from fires is used and hence the inter-annual variability of the emissions is ignored.").

**General comments:**

1. The argument in the Discussion section (Pg 11 Lines 3-6) for saturation of the aerosol radiative effect on clouds for the high fire emission years and locations (as opposed to the steady mean emissions) seems right, but the test of this hypothesis, illustrated with Figure 9, is not as convincing. Figure 9 shows how the sublinear response of the indirect effect to increases in aerosols is evident for a few grid points in a region, but does not give reason to think that the emissions vs. RFP relationship at these three grid points should apply generally. Especially give that two of the five points selected do not show the same relationship. And since the five points in the figure were already chosen specially because they had the largest FMEAN-F{yyyy} difference (among the other criteria), the figure does not show that the saturation effect matters here necessarily.

What if you were to plot timeseries (maybe monthly?) of Boreal Asia region average OC emissions, CCN (I think there is a standard CCN 0.2%SS variable output in CAM5?) and RFP, or some metric of forcing efficiency, with each timeseries standardized so they could be plotted together. Just one F{yyyy} could be plotted with FMEAN I think. This could show whether peaks and valleys in the interannual varying OC/CCN timeseries are matched by similar increases and decreases in RFP or if this saturates at the high extremes as would be expected. I am not sure if this effect would be clearly visible on a regional average but this might be something to try.

I also suggest including a standard reference for the saturation effect (maybe at Pg 11, Line 4) such as Boucher and Pham (2002). And since the first indirect effect is not isolated from all aerosol/cloud effects in this study, i.e. lifetime effect, semi-direct effect, it would be helpful to include discussion of how the additional aerosol effects factor in to estimates of the saturation effect. Or how they cloud the conclusions that can be drawn.

Thank you for your suggestions, including that of plotting a timeseries. After considering this suggestion, we have decided not to include a monthly timeseries since this would be complicated by the fact that the RFP is highly dependent on solar insolation which exhibits a strong seasonal cycle over Boreal Asia. In order to address the question of generality, we have instead opted to plot a further figure of cloud shortwave RFP versus surface organic carbon aerosol

concentration averaged across the whole Boreal Asia region for July and August. We have included this as a new Fig. S13 to complement Fig. 9, and have updated the text as follows:

"It is worth noting that the results presented in Fig. 9 have two obvious limitations. First, as pointed out above, surface organic carbon aerosol concentration may not always be a good proxy for the cloud condensation nuclei available to clouds. Second, these results may not apply generally to the Boreal Asia region because they represent only a relatively small sample of location–month combinations, selected because they have large "conventional" − "revised" differences.

"In order to partially address the second of these limitations, corresponding results showing cloud shortwave RFP versus surface organic carbon aerosol concentration averaged across the whole Boreal Asia region during the months of July and August are shown in Fig. S13. The first two observations about Fig. 9 listed above also apply to the Boreal Asia regional averages shown in Fig. S13: the surface organic carbon aerosol concentration is very similar between the "conventional" and "revised" approaches; and a logarithmic fit works well, with the cloud shortwave RFP scaling approximately linearly with the logarithm of the surface organic carbon aerosol concentration. However, the "conventional" approach still produces stronger cloud shortwave RFPs than would be expected from the regional average organic carbon aerosol concentrations, although this may be primarily due to the fact that the use of regional averages obscures differing spatial inhomogeneities in surface organic carbon aerosol concentration.

"Fig. 9 (especially Fig. 9c) and Fig. S13 provide some evidence in support of our hypothesis that the overestimation of cloud shortwave RFP strength occurs due to the non-linear influence of aerosols on clouds. In order to test this hypothesis more conclusively, it would be advantageous to perform idealised simulations designed to isolate the contribution of different aerosol sources, aerosol species, and aerosol effects. Such further analysis is outside the scope of the present study."

Thank you for suggesting that we include a reference for the saturation effect. We have added an additional sentence before the one in which we suggest that the indirect effects may saturate: "For example, there is evidence of a "sublinear dependence of cloud droplet concentrations on aerosol number" (Stevens and Feingold, 2009)."

2. Related to this, I think it would be helpful to include more details about how aerosols interact with the radiative budget in this model setup, probably in the Methods section 2.1. For example, mentioning the different effects that are included in the RFP such as the direct effect, semi-direct effect, etc., explaining how aerosols can perturb the radiative flux in this model setup. This is also important to explain for the hydrological response which is discussed in Section 3.2 and shown in Figure 8. Aerosol impacts on both stratiform and convective clouds are included even though aerosols in the model do not directly interact with the convective scheme (Section 2.1). So here is another instance where explaining how the aerosols indirectly impact convection in the methods would be helpful.

We have adopted your suggestion, and have modified the paragraph in question to include the following: "Aerosol direct and semi-direct effects are included via coupling between the aerosols and the radiation. Aerosol indirect effects on stratiform clouds are included via coupling between the aerosols and the stratiform cloud microphysics (Morrison and Gettelman, 2008; Gettelman et al., 2010). As a result of these indirect effects (Ghan et al., 2012), CESM1-CAM5 produces a relatively strong total aerosol radiative effect compared to many other global climate models (Table 7 of Shindell et al., 2013). CAM5 does not include a representation of aerosol indirect effects on convective cloud microphysics. However, the aerosols may indirectly impact convection via interaction with the radiation."

**Minor comments:**

Pg 5, Lines 15-17: There are not many estimates of fire RF but you could include Chuang et al. (2002), -1.16 Wm-2 (first indirect effect only), and Ward et al. (2012), -1.38 Wm-2 (also CAM5), as additional points of comparison.

Thank you for pointing out these two studies. We have now included them: "The global mean net RFP of $-1.3\,\mathrm{W\,m^{-2}}$ is comparable to that found by Clark et al. (2015, their "interpolation method" simulation) who also used CAM5. This value, which is dominated by the shortwave (see below), is also comparable to two previous estimates of the shortwave forcing associated with fire aerosols: an estimate of $-1.4\,\mathrm{W\,m^{-2}}$ by Ward et al. (2012, their Table 5) who also used CAM5; and an estimate of $-1.2\,\mathrm{W\,m^{-2}}$ by Chuang et al. (2002) who used CCM1/GRANTOUR to quantify the cloud albedo effect only."

Pg 6, Line 9: The foundation for the rest of the results section is developed very nicely in this section. For this line, please include how you define statistical significance here.

We have now added mention of the method used to test statistical significance, and refer readers to the table caption for details: "A "conventional" − "revised" difference of $-0.24\,\mathrm{W\,m^{-2}}$ is found (Table 1), primarily due to the cloud shortwave RFP component (Fig. 3a). Welch's $t$-test reveals that this difference is statistically significant (see Table 1 caption). These results indicate that the conventional approach of using inter-annually invariant fire aerosol emissions leads to a 23 % overestimation of the negative RFP exerted by fire aerosols on the climate system (Fig. 5a)."

Pg 6, Lines 16-17: It is mentioned here and shown in Figure 5 that fire aerosol impacts on forcing extend over ocean but it seems like only land area is considered in the regional estimates? In Ward et al. (2012) we also found large forcings ($< $ -5 Wm-2) over ocean for fire aerosols, in fact, globally the largest indirect aerosol forcings were for the marine stratocumulus decks off the coasts of western Africa and South America.

If forcings over ocean are not considered here, can you discuss how the answers might be different if these areas were included?

We have now modified the paragraph in question: "Although the discussion below focuses on land regions, readers should note that the radiative effects of

the fire aerosols are not limited to land regions but extend over ocean regions downwind of fire sources. Net RFPs stronger than $-5\,\mathrm{W\,m^{-2}}$ occur over parts of the North Pacific, Tropical Pacific, and Tropical Atlantic oceans, including the stratocumulus decks off the coasts of western South America and western Africa (Fig. 6a,b). Ward et al. (2012) similarly identified strong cloud forcing associated with these stratocumulus decks. The presence of such highly inhomogeneous forcing over ocean may lead to perturbed surface temperature gradients which in turn may lead to large-scale circulation changes and precipitation impacts (Wang, 2015). However, changes in SST gradients are outside the scope of the present study which analyses results from prescribed-SST simulations."

Pg 7, Lines 26-27: I wonder if you can speculate why the BC emissions have a higher cloud sw forcing efficiency in these regions, and why the BC sw cloud forcing does not have a consistent sign in general when looking at all global regions. Perhaps this is a result of semi-direct effects (e.g. Sakaeda et al., 2011; Koch and Del Genio, 2010).

Following your comment, we have modified the sentences in question: "The cloud shortwave RFP is primarily driven by the organic carbon emissions (Fig. 3d). The black carbon emissions also drive a negative cloud shortwave RFP, possibly as a result of semi-direct effects leading to increased cloud water path (Fig. S14d) – Koch and Del Genio (2010) have previously highlighted that there are "several mechanisms by which absorbing aerosols may either increase or decrease cloud cover"."

Pg 8, Lines 4-6: This summary point for the South American region is clear and nicely explained. I think it is important to note again here that aerosols are only microphysically active for stratiform clouds. This could be especially important in the tropical regions.

Thank you. We have added a sentence following your suggestion: "However, if the model were to include a representation of aerosol indirect effects on convective cloud microphysics, the results might be different for these convectively active tropical regions."

Pg 9, Line 1: Why is EQAS divided into two sub-regions for this study?

We chose not to use the EQAS region due to the difficulty of distinguishing land from ocean in the Maritime Continent at the model resolution. We have modified the Fig. 1 caption to include: "Unlike the land-only GFED4.0s basis-regions, the two Maritime Continent regions both include ocean as well as land, due to the difficulty of distinguishing land from ocean at a resolution of $1.9° \times 2.5°$ over the Maritime Continent." We look at the Western Maritime Continent and the Eastern Maritime Continent separately due to our interest in these regions, especially the Western Maritime Continent.

Pg 10, Lines 9-12: This sentence left me wondering why this is the case ? that is that the fire aerosols lead to a precipitation increase. I am not sure it is feasible to tease out the reasons for this given the scope of this study, but it might be worth looking at what other changes are occurring such as changes in cloud fraction, cloud depths, CDNC, etc. that might provide some insight. Or maybe this has to do with differences in snow cover due to fire aerosols.

We have modified this paragraph to clarify that total precipitation is still

suppressed, even over the Boreal regions: "The exceptions are the two boreal regions where, although total precipitation is suppressed, large-scale precipitation is actually enhanced slightly by the fire aerosols (Fig. 7b–c). This enhancement of large-scale precipitation over the Boreal regions partially offsets the much stronger suppression of convective precipitation."

Pg 10, Lines 15-16: This is a good point, and might be a good place to note that Tosca et al. (2013) and Clark et al. (2015) do show a dynamical response in precipitation to fire aerosols using a slab ocean, although with non-interannually varying emissions. Another place these could be cited would be on/near Pg 9, Line 24.

Thank you for the suggestion. However, we think that discussing these references in this section would reduce clarity. At the end of the Discussion section, however, we do mention that "if SST feedbacks were to be included, the "conventional" − "revised" RFP differences would likely impact surface temperature gradients. Changes in surface temperature gradients are known to impact precipitation patterns (Wang, 2015)."

Pg 13, Line 8: It is great that the data from this study are made available and so easily accessible.

Thank you for the encouragement!

Thank you once again for all your insightful comments.

**2   Response to Referee #2 (A. Veira)**

Aerosol emissions from open-burning fires show a distinct inter-annual variability. This manuscript by Grandey et al. quantifies the impact of the inter-annual fire emission variability on global and regional radiative effects. The authors present results from simulations with the Community Atmosphere Model version 5 (CAM5) which compare top-of-atmosphere radiative flux pertubations for inter-annually varying and inter-annually invariant emission inventories. The application of inter-annually varying fire emissions reduces the mean net radiative aerosol effect by about 23% globally and up to 58% regionally. The simulated global changes in precipitation, however, are not significant.

The scientific topics addressed by this manuscript are within the scope of ACP and the study design is adequate for investigation of the given research questions. Overall, the paper is well structured and well written. The results are presented in a clear and understandable way, supported by neat figures and tables. The detailed discussion of the individual regional impacts appears slightly long-winded in some parts, but it provides substantial information for the overall conclusions of the study and might as well be beneficial for subsequent future studies. The reference list is comprehensive and largely reasonable. Due to the large differences in the overall fire aerosol radiative forcing between CAM5 and other global aerosol models, slightly more detailed descriptions and discussions of the used emission inventory uncertainties, the applied microphysical aerosol properties and the cloud shortwave radiative flux pertubations in the model, are desirable. In addition, I have some minor comments and suggestions.

I recommend publication in ACP if the authors sufficiently address the following comments in their revised manuscript.

Thank you for your recommendation and for your suggestions, including your comment that "slightly more detailed descriptions and discussions of the used emission inventory uncertainties, the applied microphysical aerosol properties and the cloud shortwave radiative flux perturbations in the model, are desirable" – we deal with this overarching comment in the context of your general comments and specific comments below.

**General comments:**

1. The fire emission data set GFED applied in this study is a well-established and frequently used fire emission data set and there are no objections to use it for the purpose of this study. However, fire emission estimates based on other retrieval methods (e.g. the Global Fire Assimilation System GFAS, Kaiser et al., 2012) provide emission estimates which show substantial regional emission flux differences compared to GFED (e.g. Kaiser et al., 2012 and Zhang et al., 2014). Therefore I suggest at least to mention the emission estimate uncertainties in more detail in Section 2.2 and in the discussion of the results.

Thank you for highlighting this source of uncertainty. Following your suggestion, we have added two sentences about uncertainties associated with fire emissions in the Method section: "Several fire emission inventories are available

(van der Werf et al., 2010; Wiedinmyer et al., 2011; Kaiser et al., 2012). There is uncertainty associated with these fire emission inventories – for example, Lee et al. (2016) found that differences between two of these inventories contributes to uncertainty in modelled aerosol concentration over Southeast Asia."

We have also added a sentence in the second paragraph of the Results section: "In addition to large uncertainty associated with the parameterisation of indirect effects, there is also uncertainty associated with emissions of fire aerosols."

2. Globally, the radiative flux perturbation (RFP) of more than 1 Wm-2 of both, the 'conventional' and the 'revised' fire emission approaches presented in Section 3.1.1 are surprisingly large compared to most other studies which provide top-of-atmosphere radiative forcings of fire emissions which range between -0.3 Wm-2 and +0.1 Wm-2 (for a list of references see Veira et al., 2015, Section 5, page 7188). Can you provide a more detailed explanation, why this is the case? The comparatively large values of the anthropogenic / total RFP in CAM5.1 provided by Shindell et al., 2013 (cited on P3, L16 and P5, L18 in this manuscript) do not necessarily explain the very large radiative effect of the fire aerosol emissions in particular. Might the black carbon (BC) and organic carbon (OC) atmospheric lifetime largely differ in the other models? Which refractive indices for BC and OC are used in CAM5 (for a comparison to ECHAM-HAM2, see Zhang et al., 2012)? What specific implementation in the cloud microphysics is included in CAM5, but not in the other models?

Thank you for encouraging us to consider this in more detail and for pointing out the further references. We now reference the studies by Jones et al. (2007) and Unger et al. (2010) in the Introduction: "Chuang et al. (2002), Jones et al. (2007), Unger et al. (2010), Ward et al. (2012), Clark et al. (2015) and Veira et al. (2015) all appreciated the importance of modeling the indirect effects of fire aerosols."

We considered including the results of Jones et al. (2007) in the discussion of our global mean net RFP at the start of the Results section. However, we decided not to in the end due to the fact that Jones et al. (2007) used year-1860 fire emissions as their reference rather than zero fire emissions, hence their year-2000−year-1860 net RFP of $-0.3\,\mathrm{W\,m^{-2}}$ cannot be directly compared with our FMEAN−F0 net RFP – the difference in reference scenario will likely make a large difference.

Following you comment, and a comment from Reviewer #1, we have adapted the paragraph in question: "The "conventional" global mean net RFP associated with fire aerosols is $-1.3\,\mathrm{W\,m^{-2}}$ (Table 1). Note that this RFP is relative to simulation F0 which has zero emissions of fire aerosols. If a different reference, for example year-1850 emissions of fire aerosols, were to be chosen, the global mean net RFP would be smaller in magnitude. The global mean net RFP of $-1.3\,\mathrm{W\,m^{-2}}$ is comparable to that found by Clark et al. (2015, their "interpolation method" simulation) who also used CAM5. This value, which is dominated by the shortwave (see below), is also comparable to two previous estimates of the shortwave forcing associated with fire aerosols: an estimate of $-1.4\,\mathrm{W\,m^{-2}}$ by Ward et al. (2012, their Table 5) who also used CAM5; and an estimate of $-1.2\,\mathrm{W\,m^{-2}}$ by Chuang et al. (2002) who used CCM1/GRANTOUR to quantify

the cloud albedo effect only. However, a value of $-1.3\,\mathrm{W\,m^{-2}}$ is much larger than ECHAM6-HAM2's fire aerosol net RFP of $-0.2\,\mathrm{W\,m^{-2}}$ (Veira et al., 2015). This discrepancy appears to be primarily due to differences in the parameterisation of indirect effects, especially the sensitivity of stratiform clouds to organic carbon aerosol emissions (see below). CESM-CAM5, which produces an anthropogenic aerosol year-2000−year-1850 net RFP of $-1.5\,\mathrm{W\,m^{-2}}$ (Ghan et al., 2012), is known to produce a stronger net RFP than many other global climate models (Shindell et al., 2013). In addition to large uncertainty associated with the parameterisation of indirect effects, there is also uncertainty associated with the emissions of fire aerosols."

Since the net RFP is dominated by the cloud shortwave RFP, the refractive indices of BC and OC are not directly relevant here, especially since the semi-direct effect is relatively small in CAM5. Similarly, we think that any perturbation to the BC and OC lifetimes, which are approximately 4 days in CAM5 (Liu et al., 2012; Clark et al., 2015), would be at most a second-order effect. The references for the cloud microphysics (Morrison and Gettelman, 2008; Gettelman et al., 2010) are provided in the Method section.

3. Cloud shortwave RFP is identified as major driver of the total net RFP differences between the simulations with inter-annually invariant and inter-annually varying fire emissions for many regions in the results section. Can you provide any numbers on high and low cloud cover changes in these regions? Do you see distinct differences in vertical cloud profiles? Some numbers would be helpful to further explain the non-linear saturation effects which are nicely presented in the discussion section.

Thank you for your suggestion. Rather than including cloud cover changes, we have now decided to include a figure showing changes in grid-box average cloud water path in the supplemental material (Fig. S14). We refer to this new figure a handful of times in the Results section, including in the "Global distribution" section: "For the regions discussed below, negative cloud shortwave RFPs are generally associated with increases in grid-box average liquid water path and total water path (Fig. S14). Negative cloud longwave RFPs are generally associated with decreases in ice water path (Figs. S11 and S14)."

**Specific Comments:**

Abstract:

P1, L14-15: '[...], we need to more accurately quantify the effects of aerosols [...]'. This sub-clause is a very general statement and does not provide specific outcomes of this study. Therefore the last sentence of the abstract can be shortened by pointing directly to the specific importance of the inter-annual variability of fire aerosols quantified in this study.

Following your suggestion, we have re-written the final sentence of the abstract: "In order to improve understanding of the climate system, we need to take into account the inter-annual variability of aerosol emissions."

Introduction:

P1, L21-22: The references referring to peat fires in Indonesia are slightly misleading in this context. Although the sulfur content of the biomass varies substantially, all fire categories named in line 16-17 include emissions of sulfur dioxide as the emission factor inventories by Andreae and Merlet, 2001 and Akagi et al., 2011 demonstrate.

Thank you for highlighting that this sentence is potentially misleading. We have now dropped this sentence, and have mentioned sulphate in the preceding sentence instead (see response to comment below).

P1, L22-23: Does the statement 'These aerosols have a negative impact on [. . .]' refer to sulfur dioxide emissions from fire? It should be clarified that Lelieveld et al., 2015 quantified the overall impact of biomass burning aerosols (including black carbon and organic carbon) on human health, not the human health impact of sulfur emissions from fires in particular.

Following our response to your previous comment (see above), this ambiguity has now been resolved: "Other pollutants include aerosols, containing organic carbon, black carbon, and sulphate. These aerosols have a negative impact on air quality and human health (Lelieveld et al., 2015)."

P2, L1: 'Aerosols scatter or absorb incoming sunlight'. The conjunction 'or' is misleading as both processes, scattering and absorption, apply to most atmospheric aerosol species.

We have re-written this: "Aerosols can scatter and absorb incoming sunlight".

P2, L7-8: '[. . .] many other aerosol effects on clouds have also been proposed.' Please provide examples of these 'other' effects.

Following your suggestion, we now mention two further examples: "In addition to these two indirect effects, other aerosol effects on clouds have also been proposed, including a glaciation indirect effect (Lohmann, 2002) and the convective invigoration hypothesis (Andreae et al., 2004; Rosenfeld et al., 2008). For further discussion of possible aerosol effects on clouds, readers are referred to the review papers written by Lohmann and Feichter (2005), Tao et al. (2012), and Rosenfeld et al. (2014)."

P2, L11 and L14: In the reference list, there are two references given for Wang et al., 2009. Therefore it is unclear, which of the two papers is referred to in line 11 and line 14, respectively.

The first of these contains an "et al." (Wang et al., 2009), whereas the second is a single-author paper (Wang, 2009). We appreciate that this may be confusing. However, we do not know how to make the distinction clearer while still remaining within the standard ACP reference style guidelines.

Methods:

P4, L14-18: In this paragraph, it should be explicitly stated that the black carbon and organic carbon fire emission estimates from GFED applied in this study represent an important source of uncertainty (see general comment 1).

We have adopted your suggestion. Please see our response to your first general comment.

Results:

 Can you provide numbers for the increase in snow cover? It would be interesting to see the quantitative relationship between snow cover increase and surface albedo change.

Thank you for the suggestion to look at snow cover. In the section of the Results focusing on the Boreal regions, we now write that "during the summer months of June–August, the fraction of the ground covered by snow increases from 4.0% in F0 to 5.3% in FMEAN."

P9, L19: Comparing Figure S1 to Table 1, it is interesting to see that a statistically significant overestimation of the net RFP strength is found for Central America and South-east Asia. In contrast, the Eastern Maritime Continent and Temperate North America show temporal OC emission patterns of similiar magnitude like those of Central America and Southeast Asia, respectively (Figure S1), but nevertheless no statistically significant overestimation of the net RFP is simulated for the conventional approach in these regions. Might these findings indicate that the influence of the meteorology varies significantly in these regions or is this due to the saturation effects?

This is an interesting observation. Differences in meteorology likely play a major role.

P10, L1-2: Why does black carbon suppress large-scale precipitation in such a way? If available, an analysis of vertical temperature profiles / atmospheric stability changes might be helpful for a better understanding although the discussion of these relations does of course not represent the focus of this study.

We have added a sentence stating that "It is difficult to interpret the mechanisms behind these global mean features, due to spatial inhomogeneity in the regional response (see below)."

Discussion:

P11, L1: 'Rather, across much of the globe, the "conventional" approach leads to a systematic overestimation of net RFP [.  .  .]'. Table 1 and Fig. 6c show that the statistically significant overestimation is largely limited to some specific regions. Therefore the use of the statement 'much of the globe' is inappropriate in this context.

Following your suggestion, we have replaced "across much of the globe" with "for some regions".

P12, L12: What 'other factors' do you think of? Might interaction with other, non-fire related aerosol species play a role?

Thank you for encouraging us to think further about what "other factors" might include. After considering this comment, and also Referee #1's first general comment (see above), we have decided to drop the references to "other factors".

Conclusions:

P12, L28-30: This paragraph represents a nice description of those regions, where ignoring the inter-annual variability leads to an overestimation of the net radiative effect of fire aerosols. However, as shown in Table 1, there are many other regions, where no significant differences between the two scenarios are found and this should also be stated here.

Following your comment, we have included "for some regions": "Regionally, for some regions, the "conventional" approach leads to an even larger overestimation of the strength of the net radiative effect of the fire aerosols".

P13, L3-5: At this point I suggest to provide some implications and ideas for subsequent future studies, which could be initiated by the results and conclusions presented in this paper, e.g. the quantification of daily vs. monthly fire emissions or a deeper analysis of the cloud micro-physical processes which are responsible for the found saturation effects.

Following your comment, we have added a paragraph recommending avenues for further research: "Following the findings presented in this paper, we suggest three avenues for further research. First, similar simulations could be performed using other aerosol–climate models in order to test whether the conclusions of this study apply more generally if different parameterisations of aerosol indirect effects are used. Second, idealised simulations could be performed to improve understanding of the saturation of aerosol indirect effects. Third, coupled atmosphere–ocean simulations could be performed to investigate the impact of inter-annually varying emissions on the hydrological slow response and other components of the climate system, including modes of climate variability."

All figures presented in the manuscript represent elaborated and easily readable visualizations of high quality with detailed figure captions. Figure 6 and Figure 8 are neat and self-explanatory figures, but in the final manuscript the figure size should be increased in order to improve the readability of the stippling.

Thank you! We hope to increase the figure size in the final ACP manuscript.

Thank you once again for all your helpful comments.

**References**

[revised manuscript text omitted]